# Alpha oscillations and event-related potentials reflect distinct dynamics of attribute construction and evidence accumulation in dietary decision making

Azadeh HajiHosseini[1]*, Cendri A Hutcherson[1,2]

[1]Department of Psychology, University of Toronto Scarborough, Toronto, Canada; [2]Department of Marketing, Rotman School of Management, University of Toronto, Toronto, Canada

**Abstract** How does regulatory focus alter attribute value construction (AVC) and evidence accumulation (EA)? We recorded electroencephalogram during food choices while participants responded naturally or regulated their choices by attending to health attributes or decreasing attention to taste attributes. Using a drift diffusion model, we predicted the time course of neural signals associated with AVC and EA. Results suggested that event-related potentials (ERPs) correlated with the time course of model-predicted taste-attribute signals, with no modulation by regulation. By contrast, suppression of frontal and occipital alpha power correlated with the time course of EA, tracked tastiness according to its goal relevance, and predicted individual variation in successful down-regulation of tastiness. Additionally, an earlier rise in frontal and occipital theta power represented food tastiness more strongly during regulation and predicted a weaker influence of food tastiness on behaviour. Our findings illuminate how regulation modifies the representation of attributes during the process of EA.

*For correspondence:
azadeh.haji@utoronto.ca

## Introduction

Neuroeconomic theories of decision making suggest that value-based decisions can be captured by a process involving several distinct computations, including both attribute valuation and evidence accumulation (EA; *Rangel et al., 2008*). For example, when we decide what to eat for lunch, we might consider attributes like how tasty or healthy a food is, our dieting goals, current hunger levels, and so forth. Each of these attributes constitutes evidence for or against eating the food, with different attributes given differential weight depending on a person's momentary goals. To choose, current models assume that noisy neural representations of this attribute-based evidence may serve as the input to a process of sequential EA over time until sufficient evidence has accumulated to pass a decision threshold (*Forstmann et al., 2016*; *Gold and Shadlen, 2007*).

Based on this model of choice, recent work has argued that attempts to regulate dietary behaviour may operate in part by altering the weights given to specific attributes during food choice (*Hutcherson et al., 2012*; *Sullivan et al., 2015*; *Tusche and Hutcherson, 2018*). For example, dieting goals could be accomplished either by down-regulating the influence of hedonic attributes like tastiness, up-regulating the influence of attributes like healthiness, or both. Yet the precise computational route by which this up- and down-regulation occurs, and why some people succeed or fail to regulate effectively, remains unclear. Here, we took a computational decision neuroscience approach to this problem, linking computational predictions to observed neural data, to ask two questions. First, how do the computations contributing to a choice change when people regulate their

behaviour by increasing or decreasing the influence of different attributes? Second, which of these changes predict success or failure in altering food consumption?

Sequential accumulation models provide a behaviourally and biologically plausible approach to study these questions (*Forstmann et al., 2016*) that appears to capture patterns of choice, response time, and neural activity with remarkable accuracy. For example, in the perceptual decision-making domain, electroencephalogram (EEG) studies have shown that decrements in the power of alpha (9–12 Hz) and theta (4–8 Hz) oscillations correlate with the predicted build-up of evidence (*van Vugt et al., 2012*; *Werkle-Bergner et al., 2014*). Analyses of event-related potentials (ERPs) also show that the P300 component elicited by oddball stimuli in auditory and visual paradigms satisfies the characteristics of an EA signal (decision variable) (*O'Connell et al., 2012*; *Twomey et al., 2015*). Parallel to this, in the value-based decision-making domain, EEG and MEG studies have found that model predictions about the temporal dynamics of EA during decisions correlate with both parietal and frontal theta and alpha (*Hunt et al., 2012*), and with beta (18–20 Hz) and gamma (40–80 Hz) oscillations (*Polanía et al., 2014*). Given that value-based decision making engages multiple cognitive processes and involves communication across different regions, these diverse results are perhaps not surprising. However, they raise important questions about what precise computations these EEG dynamics represent, and whether and how these signals might be altered by up- and down-regulating the influence of food attributes on choice.

These questions are vital for understanding the underlying regulatory mechanisms in value-based decision making since it could target multiple distinct phases of the EA process. Although considerable evidence suggests that self-regulation during decision making modifies choices by changing the influence of attributes on behaviour (*Hare et al., 2009*; *Hare et al., 2011*; *Hutcherson et al., 2012*; *Sullivan et al., 2015*; *Tusche and Hutcherson, 2018*), it is unclear how such changes are accomplished neurally. For example, observed decreases in the influence of food tastiness on behaviour could be accomplished either by abolishing attribute representations of tastiness or by preventing the incorporation of such representations into the EA process. If the former is true, regulation should directly influence the neural representation of an attribute such as tastiness (perhaps early on in the decision process), whereas if the latter is true, the neural representation of an attribute might remain intact and only the later stages of decision making, when the attribute is incorporated into EA, should change. Unfortunately, fMRI studies (*Hare et al., 2009*; *Hutcherson et al., 2012*) cannot dissociate the neural representations of attributes and EA with the necessary temporal resolution for testing these distinct processes.

EEG studies provide some clues in this regard, although they also leave open a number of questions. For example, some research suggests that when instructed to attend to food healthiness, the amplitude of a fronto-central ERP component around 300–500 ms post-food stimulus correlated more strongly with food healthiness (*Harris et al., 2013*). However, it is unclear whether this ERP component represents changes to early attribute representations or changes to their subsequent weight in the EA process. Moreover, while regulation reduced *behavioural* sensitivity to *tastiness* considerations, ERP amplitude in the same time window correlated with tastiness regardless of regulatory effort (*Harris et al., 2013*). This leaves open the question of how the influence of *tastiness* on behaviour was suppressed.

Given previous research that clearly links oscillatory dynamics to EA (*Hunt et al., 2012*; *Polanía et al., 2014*; *van Vugt et al., 2012*; *Werkle-Bergner et al., 2014*), and more ambiguous evidence linking value representations to fronto-central ERPs (*Harris et al., 2013*), we took a model-based approach to determine whether part of the answer to these questions might lie in the functional role of oscillations in incorporating attribute values into the EA process. More specifically, we asked subjects to make food choices naturally or under two regulation conditions. In the HEALTH condition, designed to increase the influence of the healthiness attribute and, secondarily, decrease the influence of the tastiness attribute, we asked subjects to explicitly focus on healthy eating. In the DECREASE condition, designed to decrease the influence of the *tastiness* attribute on choice without strongly modifying the influence of the healthiness attribute, we asked participants to decrease their desire and craving for foods generally.

We used a computational model of choice to first fit the behavioural data, and then simulated the predicted dynamics of both attribute representation and EA during food choice under the three conditions. These simulations suggested that the computations associated with attribute values and EA follow distinct temporal trajectories. Then, we identified both ERP and oscillatory correlates of

tastiness and healthiness representations during self-regulation and asked whether and how the time course of these correlates resembled the simulated computational dynamics of both attribute representation and EA during natural and regulated choices. Finally, we asked whether and how changes in these oscillatory dynamics correlated with individual differences in regulatory success, as measured by changes in model parameters. Our results suggested that the time course of ERPs correlated more strongly with model-based predictions for attribute representations, while suppression of alpha oscillatory activity correlated better with model-based predictions for EA signals. Importantly, regulation had a larger effect on the alpha-correlate of tastiness compared to the ERP-correlate of tastiness, and this effect appeared to be mediated by an increase in the power of theta oscillations in the early stages of decision formation.

## Results

Subjects performed a food choice task (*Figure 1*) in which they decided whether or not they wanted to consume different foods, while focusing on one of three goals: respond naturally to all foods (NATURAL), focus on healthy eating (HEALTH), or focus on decreasing desire for all foods (DECREASE; see Materials and methods for a more detailed description of the task). On each trial of the choice task, subjects made one of four responses (Strong No, No, Yes, or Strong Yes) to indicate whether they wanted to eat the food that was displayed on the screen. After the choice task, subjects rated each food stimulus for its *tastiness* and *healthiness* on a 1–6 scale, allowing us to examine how these attributes were encoded in neural activation at the time of choice.

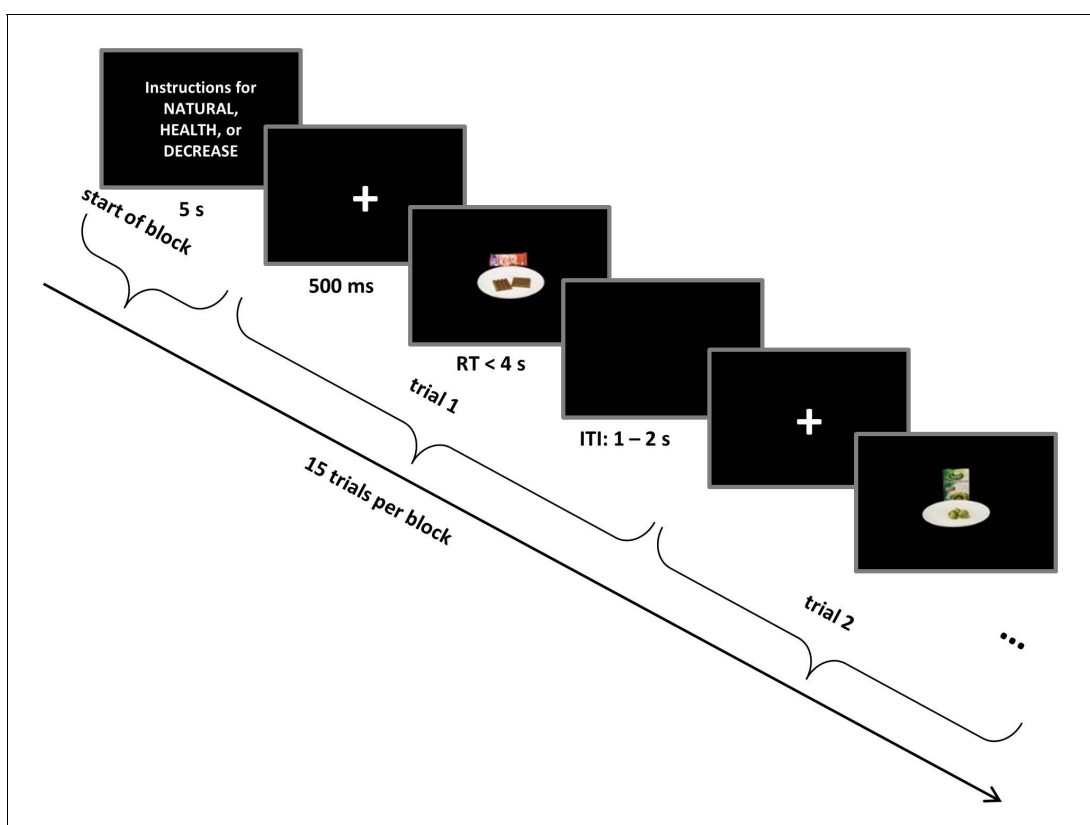

**Figure 1.** Self-regulation choice task structure. Trials occurred in interleaved blocks of NATURAL, HEALTH (designed to increase the influence of healthiness on choice), and DECREASE (designed to decrease the influence of tastiness on choice) conditions. Each block started with instructions related to NATURAL or regulation conditions (HEALTH or DECREASE) that remained on the screen for 5 s followed by 15 choice trials. Every food stimulus was preceded by a fixation cross that was colour-coded for the condition. On each trial, subjects made a response (Strong No, No, Yes, or Strong Yes) to indicate whether they wanted to eat the food presented on the screen. See Materials and methods for a detailed description of the task and conditions.

## Effects of regulatory focus on behaviour

We first sought to confirm that regulation influenced choice behaviour, focusing first on overall food acceptance rates. A three-way ANOVA with condition (NATURAL, HEALTH, DECREASE) as a within-subjects factor revealed differences in acceptance rate across conditions ($F_{(2,98)}$ = 18.73, p<0.001). As expected, subjects accepted foods less often in DECREASE compared to both NATURAL ($t_{(49)}$ = −4.0, p<0.001) and HEALTH ($t_{(49)}$ = −3.9, p<0.001), which did not differ from each other (p>0.05). Response times were also affected by regulation ($F_{(2,98)}$ = 7.33, p=0.004). Subjects were faster in the NATURAL compared to both HEALTH ($t_{(49)}$ = −5.7, p<0.001) and DECREASE ($t_{(49)}$ = −2.6, p=0.01) which were not significantly different from each other (p>0.05).

## Effects of regulatory focus on computational model parameters

To examine the computational bases of these effects, we estimated a drift-diffusion model (DDM) with six parameters: weights on tastiness ($w_{tastiness}$) and healthiness ($w_{healthiness}$), a generic value constant (*ValConst*; that is, a general tendency to assign positive or negative values to foods, unrelated to tastiness or healthiness), threshold (*trs*), non-decision time (*ndt*), and starting point bias (*spbias*; see Materials and methods for details; *Figure 2* and *Figure 2—figure supplement 1*).

As expected, we found that regulation had a significant effect on the weight given to tastiness ($w_{tastiness}$: $F_{(2,98)}$ = 33.17, p<0.001) and healthiness ($w_{healthiness}$: $F_{(2,98)}$ = 62.50, p<0.001) and on the value constant ($F_{(2,98)}$ = 22.45, p<0.001). Post-hoc paired t-tests showed that weight on tastiness decreased to a similar extent in HEALTH ($t_{(49)}$ = −7.42, p<0.001) and DECREASE ($t_{(49)}$ = −6.18, p<0.001) compared to NATURAL. By contrast, while the weight on healthiness ($w_{healthiness}$) significantly increased in both HEALTH and DECREASE compared to NATURAL (HEALTH: $t_{(49)}$ = 9.38, p<0.001; DECREASE: $t_{(49)}$ = 3.56, p<0.001), it was significantly larger in HEALTH compared to DECREASE ($t_{(49)}$ = 7.12, p<0.001). In addition, the value constant (*ValConst*) was more negative in DECREASE compared to NATURAL ($t_{(49)}$ = −4.11, p<0.001) and HEALTH ($t_{(49)}$ = −6.09, p<0.001), and more negative in NATURAL compared to HEALTH ($t_{(49)}$ = −2.38, p=0.02). Thus, while both strategies decreased the influence of tastiness on choice, HEALTH focus specifically increased the influence of healthiness, and DECREASE resulted in a decrease in value that is non-specific to tastiness or healthiness.

Regulation also influenced other parameters of the model beyond attribute weights. In particular, it changed the decision threshold (*trs*: $F_{(2,98)}$ = 25.45, p<0.001) and non-decision time (*ndt*: $F_{(2,98)}$ = 4.69, p=0.01). Notably, we found little evidence that regulation affected starting point biases (*spbias*: p>0.05), suggesting that most of the effects of regulation occurred during post-stimulus processing rather than in pre-stimulus preparation. Post-hoc paired t-tests on significant effects showed that decision threshold was higher in DECREASE compared to NATURAL ($t_{(49)}$ = 6.0, p<0.001) and HEALTH ($t_{(49)}$ = 3.67, p<0.001) and in HEALTH compared to NATURAL ($t_{(49)}$ = 4.87, p<0.001). Non-decision time was also shorter in DECREASE compared to NATURAL ($t_{(49)}$ = −2.32, p=0.02) and HEALTH ($t_{(49)}$ = −2.72, p=0.009) but was not different in HEALTH compared to NATURAL (p>0.05). These results indicate that regulation more strongly changes value components as opposed to creating a value-independent bias.

## Model simulations of expected EA dynamics

Before turning to the neural results, we sought to use the best-fitting model parameters to simulate the expected time course of three neural signals related to evidence and EA: the expected time course of a signal specifically encoding tastiness information (taste-AC), the expected time course of a signal specifically encoding healthiness information (health-AC), and the expected time course of a signal related to the EA process itself, which accumulates and integrates information about both tastiness and healthiness (*Figure 2*; see Materials and methods for details). We then asked how trial-by-trial variation in tastiness and healthiness ratings would correlate over time with the magnitude of each of these signals, over and above RT (Model 1: $taste/health - AC\ or\ EA(t) \cong b(t)_0 + b(t)_{tastiness}*tastiness + b(t)_{healthiness}*healthiness + b(t)_{rt}*rt$). This analysis essentially replicated in simulated data the time-course analyses we also applied to the observed EEG data, and therefore provided a hypothesized time course for signals to look for in the EEG data.

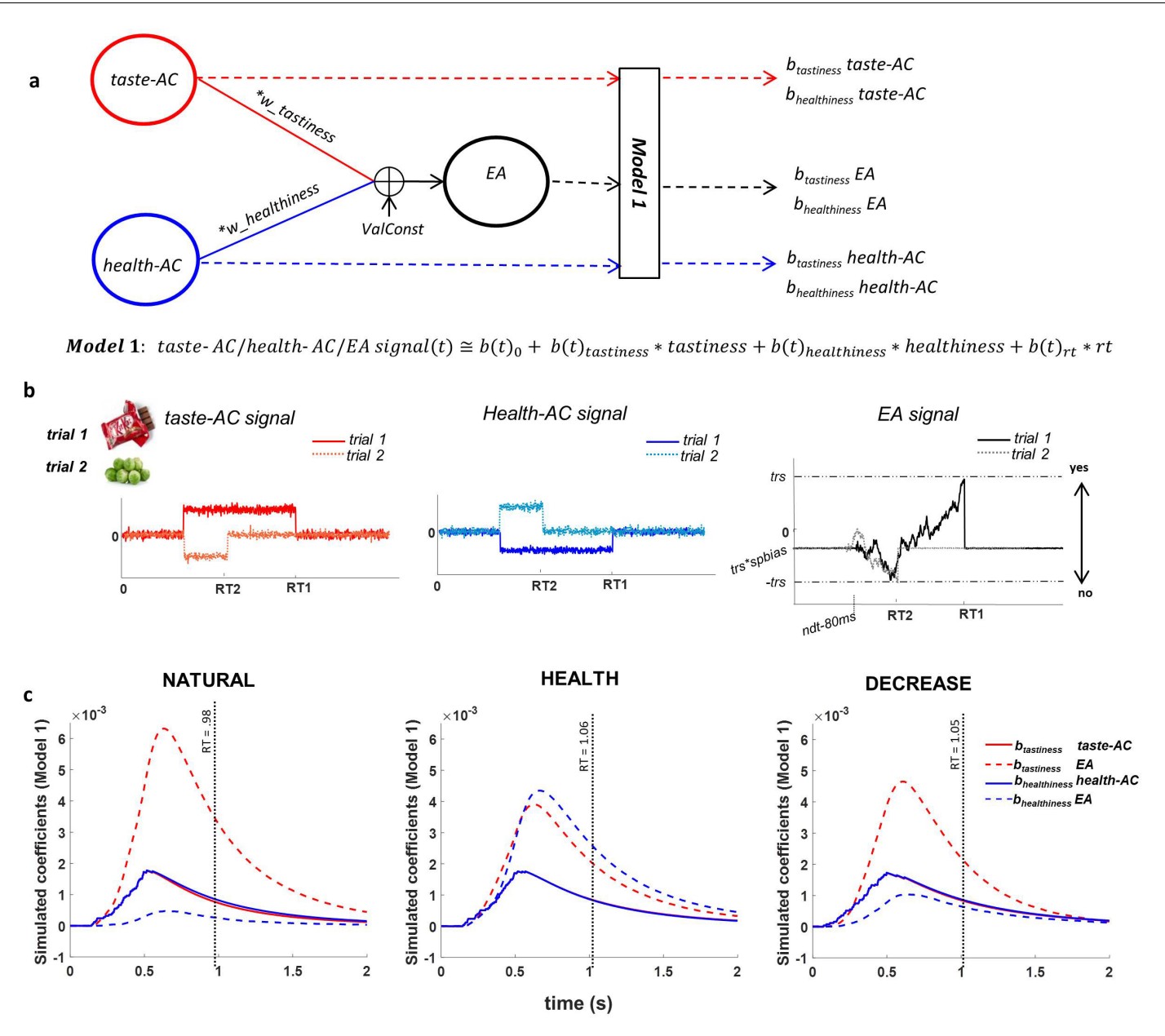

**Figure 2.** Attribute construction (AC) and evidence accumulation (EA) models. (a) AC and EA representations based on the drift-diffusion model (DDM) structure; we assumed that taste attribute construction (taste-AC) and health attribute construction (health-AC) signals in each trial integrated based on hierarchical multi-attribute drift diffusion model (HDDM) drift rates (drift (t) = $w_{tastiness}$ * tastiness (t) + $w_{healthiness}$ * healthiness (t) + ValConst) and created the EA signal based on other DDM parameters. Model 1 computes the association of individually perceived tastiness and healthiness of each food with each of the taste-AC, health-AC, and EA signals. (b) Examples of time course of taste-AC (red), health-AC (blue), and EA (black) signals in two trials where a tasty but unhealthy (trial 1, solid lines) and a healthy but non-tasty (trial 2, dotted lines) food were presented. RT1 and RT2 are the reaction times on trial 1 and trial 2. Parameters used for these simulation examples are $w_{tastiness}$ = 0.7, $w_{healthiness}$ = 0.1, ValConst = 0.5, threshold = 0.9, spbias = 0.45, nondec = 0.462 s. (c) Model 1 coefficients showing the association between food tastiness ratings and the simulated taste-AC signal (solid red line), food tastiness ratings and the simulated EA signal (dashed red line), food healthiness ratings and the simulated health-AC signal (solid blue line), and food healthiness ratings and the simulated EA signal (dashed blue line) using individual HDDM parameters in NATURAL, HEALTH, and DECREASE conditions. Coefficients are averaged across subjects. Average RT in each condition is shown on the x-axes (mean RT = 0.98 s, 1.06 s, 1.05 s for NATURAL, HEALTH, and DECREASE, respectively). See *Figure 2—figure supplement 1* and *Figure 2—source data 1* for fitted model parameters. The online version of this article includes the following source data and figure supplement(s) for figure 2:

**Source data 1.** Group hierarchical multi-attribute drift diffusion model (HDDM) parameter estimates for each condition.

*Figure 2 continued on next page*

*Figure 2 continued*

**Figure supplement 1.** Behaviour and drift-diffusion model (DDM) parameters and distributions; averaged (**a**) acceptance rate and (**b**) reaction time (RT) are shown.

Importantly, we observed a characteristic profile that distinguished simulated neural signals primarily related to attributes from neural signals primarily related to the EA process. Across the three conditions, the average association between food tastiness ratings and the simulated taste attribute signal ($b_{tastiness}$ taste-AC) reached a maximum at 516 ms, significantly earlier than the corresponding peak of average association between tastiness ratings and the EA signal ($b_{tastiness}$ taste-EA) at 630 ms (t(49) = −23.64, p<0.001; *Figure 2c*). Similarly, the average association between food healthiness ratings and the healthiness attribute signal ($b_{healthiness}$ health-AC) reached a maximum at 568 ms, significantly earlier than the peak of average contribution of food healthiness to the EA signal ($b_{healthiness}$ health-EA) at 662 ms (t(49) = −7.07, p<0.001; *Figure 2c*). These differences in the time course of model simulations suggest that signals related to attribute representations would be expected to peak around 516–570 ms, while signals related to EA would be expected to peak later, around 630–660 ms.

## ERP-correlates of model-predicted signals

Having identified putative time courses for both tastiness and healthiness attribute representations, as well as EA signals, we began by comparing model-simulated results against observed single-trial ERPs time-locked to food presentation (see Materials and methods). Using Model 2, in which we regressed the trial-level ERP signal at each time point on tastiness and healthiness (i.e. $ERP(t) \cong b(t)_0 + b(t)_{tastiness}*tastiness + b(t)_{healthiness}*healthiness + b(t)_{rt}*rt$), we asked whether there were any ERP signatures that correlated with taste-AC or taste-EA when no self-regulation was applied, and whether self-regulation in HEALTH and DECREASE conditions modulated these signatures. Based on previous results (*Harris et al., 2011*; *Harris et al., 2013*), we expected to observe a fronto-central ERP component that correlated with tastiness and healthiness, and sought to determine whether this component might match the predicted dynamics of attribute valuation or EA.

### ERPs correlate with the simulated influence of tastiness on the taste-attribute signal

Based on our significance criteria (see Materials and methods for details), we found that ERPs in fronto-central (t(49) = 3.45, p=0.004) channels were correlated with tastiness in the NATURAL condition. We refer to this as the *ERP-correlate of tastiness* (*Figure 3a*). However, this effect showed no significant differentiation between NATURAL and regulated conditions (p>0.05; but see *Figure 3c* for other differences in the scalp maps across NATURAL and DECREASE). The time course of the ERP-correlate of tastiness was closely predicted by the simulated influence of tastiness on the taste-AC signals (r = 0.89, p<0.001) and, somewhat less clearly but still significantly, the taste-EA signal (r = 0.77, p<0.001). It was also a significantly closer match for the taste-AC than the taste-EA signal (z = 2.55, p=0.005; *Figure 3b*). The magnitude of the ERP-correlate of tastiness was not correlated with weight on taste in the NATURAL condition or behavioural regulation (i.e. the difference in the weight on taste between regulation conditions) across subjects. See *Figure 3—figure supplement 1d, e* for the parietal component of ERP-correlate of tastiness. We also looked for signals matching the healthiness attribute signal, but there were no ERP signals that met our joint criterion for healthiness (i.e. correlated significantly with model-predicted time courses for healthiness and independently correlated with trial-by-trial variation in food healthiness).

### Early ERPs correlate with healthiness when subjects focus on health

Because we failed to find ERP signals for healthiness that matched our model-predicted time course, we also conducted an exploratory (i.e. not model-based) analysis of ERPs, which revealed that trial-by-trial ERPs in the HEALTH condition were correlated significantly with healthiness ~200–400 ms post-food in occipital channels (t(49) = 3.12, p=0.007; *Figure 3—figure supplement 1a*). We refer to this as the *ERP-correlate of healthiness* (see *Figure 3—figure supplement 1c* for scalp distribution and *Figure 3—figure supplement 1b* for frontal component time course). However, this early

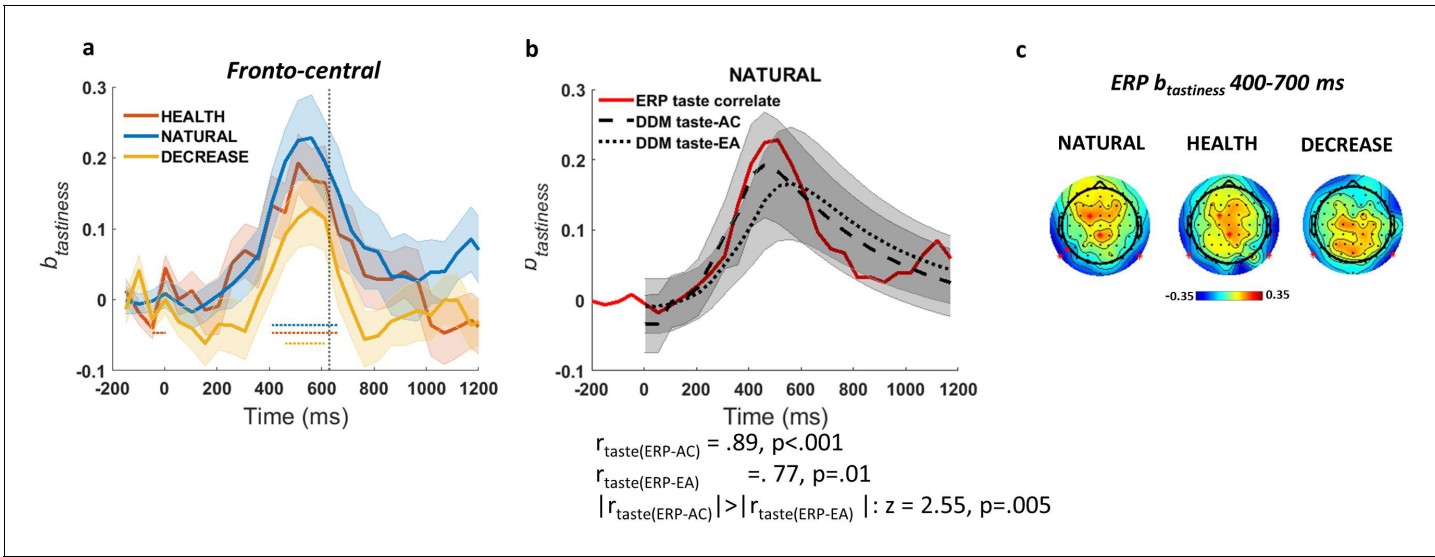

**Figure 3.** Event-related potential (ERP)-correlate of tastiness. (**a**) ERP coefficients for tastiness in Model 2 in NATURAL condition satisfy significance criteria (see Materials and methods) on fronto-central channels ~400–700 ms post-food. (**b**) The time course of ERP-correlate of tastiness (red line) is correlated with time course of contribution of tastiness to taste attribute construction (taste-AC; dashed black line) more than evidence accumulation (taste-EA; dotted black line) signal (Model 1, *Figure 2c*). (**c**) Scalp distribution of the ERP-correlate of tastiness in NATURAL, HEALTH, and DECREASE is shown. Shaded error bars show within-subject standard error of the mean. Horizontal dotted lines show significant time bins (p<0.05). See *Figure 3—figure supplement 1* for the ERP-correlate of healthiness and the parietal component of the ERP-correlate of tastiness.

The online version of this article includes the following figure supplement(s) for figure 3:

**Figure supplement 1.** Event-related potential (ERP)-correlate of healthiness and parietal component of the ERP-correlate of tastiness.

ERP signature of healthiness did not correlate with the behavioural increase in healthiness weight across subjects. We thus do not consider it further here.

## Time-frequency power correlates of model-predicted signals

To compare the model-simulated results against observed oscillatory neural responses, we extracted trial-by-trial event-related changes in power of all oscillatory ranges from 200 ms before to 1000 ms after the presentation of the food stimulus. We then regressed averaged power in each trial on trial-by-trial tastiness ratings (Model 2; $tfPower(t) \cong b(t)_0 + b(t)_{tastiness}*tastiness + b(t)_{healthiness}*healthiness + b(t)_{rt}*rt$), separately for each of 25 time bins beginning 200 ms prior to food onset and ending 1000 ms after food presentation (duration ~49 ms). We asked whether there were any oscillatory signatures that correlated with the model-predicted taste-AC or taste-EA signal when no self-regulation was applied, and whether self-regulation in HEALTH and DECREASE conditions modulated these signatures.

### Alpha power correlates with the simulated influence of tastiness on EA

Based on our significance criteria (see Materials and methods for details), we identified an alpha power signature showing a regulation-modulated sensitivity to tastiness that was in line with model predictions (*Figure 4a*). In particular, in the NATURAL condition, we found a significant change in alpha power over left-frontal and right parietal-occipital channels (*Figure 4c*, top left) that negatively correlated with tastiness ratings from 500 to 1000 ms after presentation of food (t(49) = −3.78, p<0.001). By contrast, alpha power during the two regulation conditions (HEALTH and DECREASE) was not significantly associated with tastiness. The difference in correlation between tastiness and alpha power in the window from 500 to 1000 ms was significant for the contrast of DECREASE vs. NATURAL (t (49) = 2.40, p=0.02) and marginally significant for HEALTH vs. NATURAL (t (49) = 2.01, p=0.05; *Figure 4c*, bottom). For simplicity, we call this effect the *alpha-correlate of tastiness*.

We further asked whether changes in the *alpha-correlate of tastiness* in regulation conditions compared to NATURAL predicted individual differences in DDM tastiness weight changes ($\Delta w_{tastiness}$) in regulation compared to NATURAL, as would be expected if this signal represents a

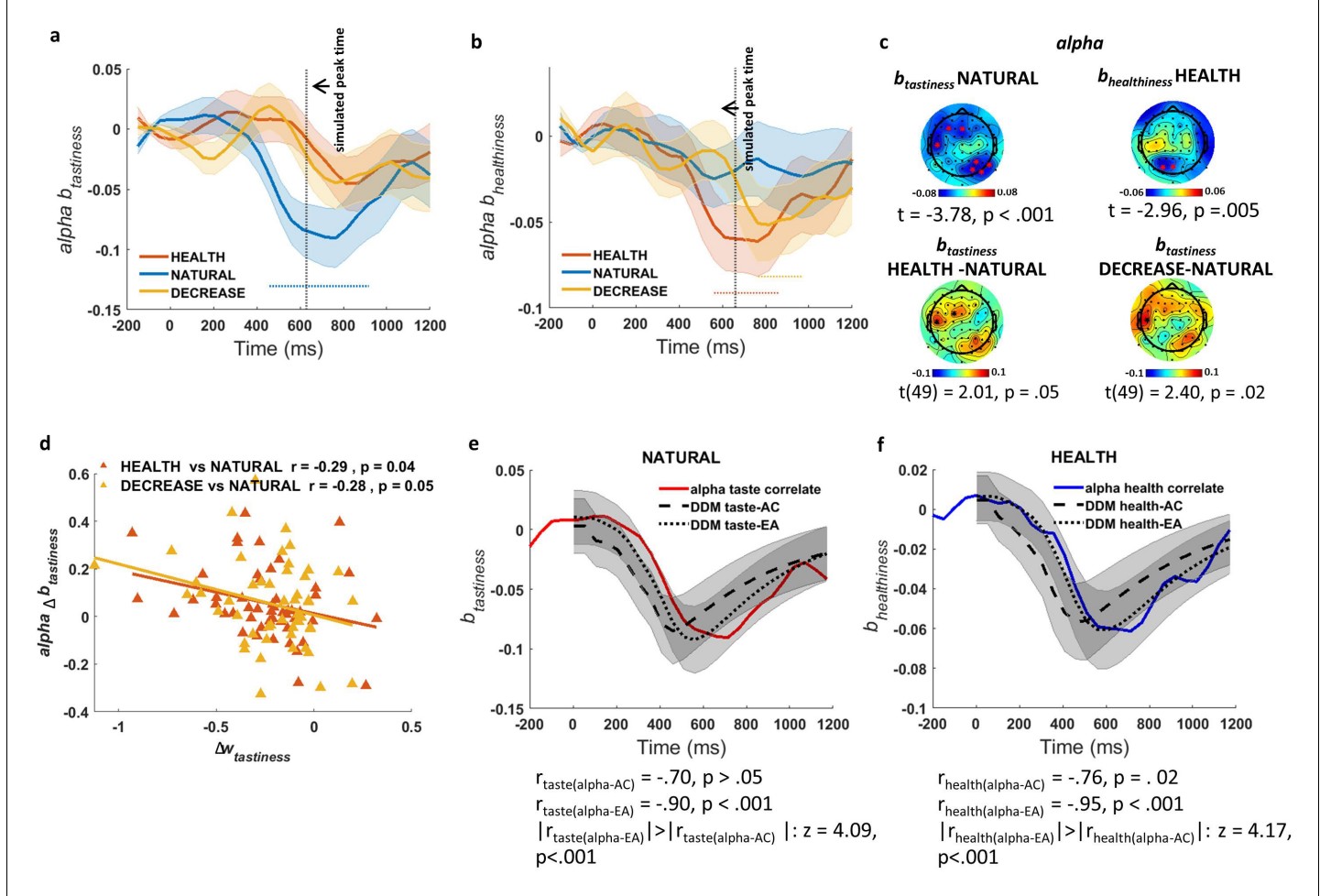

**Figure 4.** Association of alpha power with tastiness and healthiness. (a) Alpha-correlate of tastiness: alpha power coefficients for tastiness in Model 2 in NATURAL condition satisfy significance criteria (see Materials and methods) on frontal and occipital-parietal channels ~500–1000 ms post-food. (b) Alpha-correlate of healthiness: alpha power coefficients for healthiness in Model 2 in HEALTH condition satisfy significance criteria (see Materials and methods) on frontal and occipital channels ~500–1000 ms post-food. (c) Scalp distribution of the alpha-correlates of tastiness (top left) and healthiness (top right) and the difference in *alpha-correlate of tastiness* in the HEALTH vs. NATURAL (bottom left) and DECREASE vs. NATURAL (bottom right) are shown. (d) Alpha-correlate of tastiness predicts successful down-regulation of tastiness influence (w$_{tastiness}$) across subjects. (e) Time course of the *alpha-correlate of tastiness* (red line) is correlated with time course of contribution of tastiness to taste attribute construction (taste-AC; dashed black line) and evidence accumulation (taste-EA dotted black line) signals (Model 1, *Figure 2c*). (f) Time course of *alpha-correlate of healthiness* (blue line) is correlated with time course of contribution of healthiness to health attribute construction (health-AC; dashed black line) and evidence accumulation (health-EA; dotted black line) signals (Model 1, *Figure 2c*); shaded error bars show within-subject standard error of the mean. Horizontal dotted lines show significant time bins (p<0.05). See *Figure 4—figure supplements 1–3* for alpha-correlates of tastiness and healthiness calculated separately for faster and slower reaction times. *Figure 4—figure supplement 4* also shows the time-frequency maps of averaged power for food- and response-locked data.

The online version of this article includes the following figure supplement(s) for figure 4:

**Figure supplement 1.** Alpha-correlates of tastiness and healthiness for food-locked and response-locked data for fast and slow trials; *alpha-correlate of tastiness* shown separately for trials with (a) fast and (b) slow responses in food-locked data.

**Figure supplement 2.** Time course of averaged (a) food-locked and (b) response-locked alpha power in fast, slow, and medium trials in the NATURAL condition.

**Figure supplement 3.** Alpha-correlates of tastiness and healthiness (b$_{tastiness}$ and b$_{healthiness}$) for food-locked and response-locked data in the taste-sensitive channels in NATURAL and health-sensitive channels in HEALTH condition shown separately for fast and slow trials.

**Figure supplement 4.** Time-frequency plots of averaged channels (top), Fz (middle), and Pz (bottom) in (a) food- and (b) response-locked electroencephalogram.

computation directly related to choice. As hypothesized, we found a correlation between the decrease in the weight on tastiness ($\Delta w_{tastiness}$) and the decrease in the *alpha-correlate of tastiness* ~500–1000 ms post-food, both for the comparison of DECREASE vs. NATURAL (Pearson r = −0.28, p=0.05) and the comparison of HEALTH vs. NATURAL (Pearson r = −0.29, p=0.04) (*Figure 4d*).

Finally, we tested whether alpha power in channels where the *alpha-correlate of tastiness* was found correlated better with the taste-AC or taste-EA time courses predicted by the DDM (*Figure 2c*). Importantly, we found that the *alpha-correlate of tastiness* was predicted closely by taste-EA ($r_{alpha-EA}$ = −0.90, p<0.001) but not taste-AC ($r_{alpha-tastiness}$ = −0.70, p>0.05), with the taste-EA signal showing a more precise relationship compared to the taste-AC signal across time (z = 4.09, p<0.001; *Figure 4e*). These results suggest that regulation may operate in part by altering alpha frequency representations of tastiness attributes, and that this modulation might occur in relationship to EA rather than attribute construction.

We also sought to use a second way of examining the extent to which this signal might better track EA or attribute construction processes. In particular, we reasoned that if this signal is related to EA, it should show different temporal dynamics as a function of response time: when responses terminate quickly, the sensitivity to tastiness in alpha signals should also terminate quickly. However, when choices take longer, sensitivity of alpha to tastiness should endure for longer since it takes longer for the EA process to terminate. Follow-up tests confirmed this hypothesis: we found that alpha signal was sensitive to tastiness for different durations on trials where the participant responded quickly compared to trials where they responded slowly (*Figure 4—figure supplement 1a, b*).

Next, we conducted a further test of the idea that the *alpha-correlate of tastiness* relates to EA by examining response-locked data in fast and slow decisions. Prior research suggests that signals related to EA build-up gradually and peak just prior to the time of response, regardless of whether that response occurs quickly or slowly (e.g. *O'Connell et al., 2012*; *Twomey et al., 2015*). We thus theorized that we should observe a gradual build-up in alpha sensitivity to tastiness that peaks just prior to the time of response for both fast and slow responses. As expected, we found that alpha signals at taste-sensitive channels (*Figure 4c*, top left) showed a build-up of correlation with tastiness in NATURAL trials that peaked just before the time of response both when decisions were made quickly and when they were made slowly (*Figure 4—figure supplement 1c, d*). Intriguingly, this analysis also revealed an additional significant correlation with healthiness in taste-sensitive channels prior to the response on slow trials in the NATURAL condition, but not on the fast trials (*Figure 4—figure supplement 3e, f*). The time courses of these signals correlated with the simulated time course of response-locked EA (r = −0.73, p>0.05), albeit failing to reach full significance based on our a priori criteria.

Finally, we asked whether the average alpha power (as opposed to correlation with tastiness) reached a fixed value just prior to the response for fast and slow RTs, as would be expected if this signal produces the response directly. Although we found that raw alpha signals in this region did build gradually, peaking in the time before response (*Figure 4—figure supplement 2*), they did not always reach a fully clear and consistent value, as might be predicted if these signals provided the final trigger for the motor response.

Taken together, these findings suggest that, while the *alpha-correlate of tastiness* shows patterns consistent with some form of EA, it may represent an intermediate stage of EA based on value that is then passed on to other mechanisms that accumulate evidence for a particular motor response. Such architecture would be consistent with observations from other work finding evidence for multiple EA mechanisms working at different stages to determine the final choice (e.g. *Steinemann et al., 2018*).

## Alpha power correlates with the simulated influence of healthiness on EA

Next, we turned to the healthiness attribute using a similar analytical approach. Notably, in the HEALTH condition, we again found significant effects restricted to alpha power in frontal and parietal-occipital channels (*Figure 4b*). Alpha power in these channels was negatively correlated with healthiness from ~500–1000 ms after presentation of food (t(49) = −2.96, p=0.005; *alpha-correlate of healthiness*; *Figure 4c*, top-right). The timing of this signal correlated with the predicted time course of the health-EA signal ($r_{alpha-EA}$ = −0.95, p<0.001) and, less so, to the health-AC signals

($r_{\text{alpha-healthiness}}$ = −0.76, p=0.02) signals. The time course was significantly more strongly correlated with the health-EA signal than the health-AC signal (z = 4.17, p<0.001; *Figure 4f*).

As might be expected from model predictions, alpha power during the NATURAL and DECREASE conditions was not significantly associated with healthiness, although the contrast of average $b_{\text{healthiness}}$ for alpha power between HEALTH and DECREASE or NATURAL in these channels ~500–1000 ms post-food failed to reach significance. Unlike the alpha-tastiness correlate, we found no evidence that the differences in alpha-healthiness correlate across conditions correlated with individual differences in the change in model parameters for the healthiness weight.

## Early theta power correlates with later suppression of tastiness influence on EA

Having shown that we observed a correspondence between model-predicted changes in attribute representations and suppression of oscillations in the alpha range, we next turned to a more exploratory analysis, similar to that conducted for ERPs. Specifically, we asked whether there were any oscillatory changes that might distinguish regulatory conditions in relation to tastiness and healthiness, but that did *not* conform to model predictions. Strikingly, and in contrast to alpha power, which tracks tastiness in the NATURAL condition but not regulation conditions, we found a single pattern matching our significance criterion: theta power at channels over occipital and frontal areas was positively correlated with tastiness early in the trial (200–500 ms) in both HEALTH (t(49) = 3.46, p=0.005; *Figure 5a, b*, top left) and DECREASE (t(49) = 2.8, p=0.01; *Figure 5a, b*, top-right) conditions independently. This effect was also larger compared to NATURAL in both HEALTH (t(49) = 2.90, p=0.009; occipital channels) and DECREASE (t(49) = 2.28, p=0.03; occipital and left central channels) (*Figure 5b*, bottom). For simplicity, we call this effect the *theta-correlate of tastiness*.

Given the early appearance of this signal compared to the *alpha-correlate of tastiness*, we speculated that it might represent a 'control' process, rather than a representation of the tastiness attribute per se. This control process might be responsible for reducing the influence of tastiness attribute representations (which we observed in ERP signals) on the EA process (observed in alpha oscillations). This interpretation predicts that, despite theta correlating with tastiness, the magnitude of this correlation should actually be *negatively* associated with regulation-induced decreases in the weight on tastiness, and that theta and alpha signatures might also be related. These predictions were confirmed. We found that subjects with greater increases in the *theta-correlate of tastiness* between 200 and 500 ms showed greater *decreases* in the weight on tastiness ($\Delta w_{\text{tastiness}}$), and this was independently true for the contrast of both HEALTH (Pearson r = –0.36, p=0.01) and DECREASE (Pearson r = –0.29, p=0.04) vs. NATURAL (*Figure 5c*). Moreover, changes in the alpha- and theta-correlates of tastiness in DECREASE vs. NATURAL were correlated across subjects (Pearson r = 0.43, p<0.001; *Figure 5d*). In other words, the *more* strongly tastiness was encoded in theta power early in the trial, the *less* strongly it was represented in the suppression of alpha power later in the trial. This relationship was nonsignificant in the HEALTH condition (Pearson r = 0.12, p=0.39; *Figure 5d*).

No other frequency ranges showed any effects of attribute or condition that met our criteria.

## Discussion

Over the last few years, research has provided ample evidence that decision makers can modulate the influence of different attributes on behaviour using cognitive regulation strategies that focus on attending to or ignoring specific attributes (e.g. *Hare et al., 2011*; *Hutcherson et al., 2012*; *Tusche and Hutcherson, 2018*). However, the neurocomputational process by which they achieve this modulation remains unclear. Using a computational model of attribute valuation and EA, we found distinct correlates of attribute construction and EA that also showed differential effects of regulation. Whereas a fronto-central ERP component matched the predicted time course of attribute-related computations and tracked the tastiness of a food regardless of regulatory focus, alpha power recorded from frontal and occipital areas of the scalp correlated more strongly with the predicted time course of EA and tracked variation in the tastiness of food only when subjects naturally decided what foods to eat: the tastier the food, the lower the alpha power. When subjects regulated their responses in a way that reduced the behavioural influence of tastiness, this function was compromised, that is, alpha no longer tracked tastiness values. Moreover, changes in alpha, but not changes

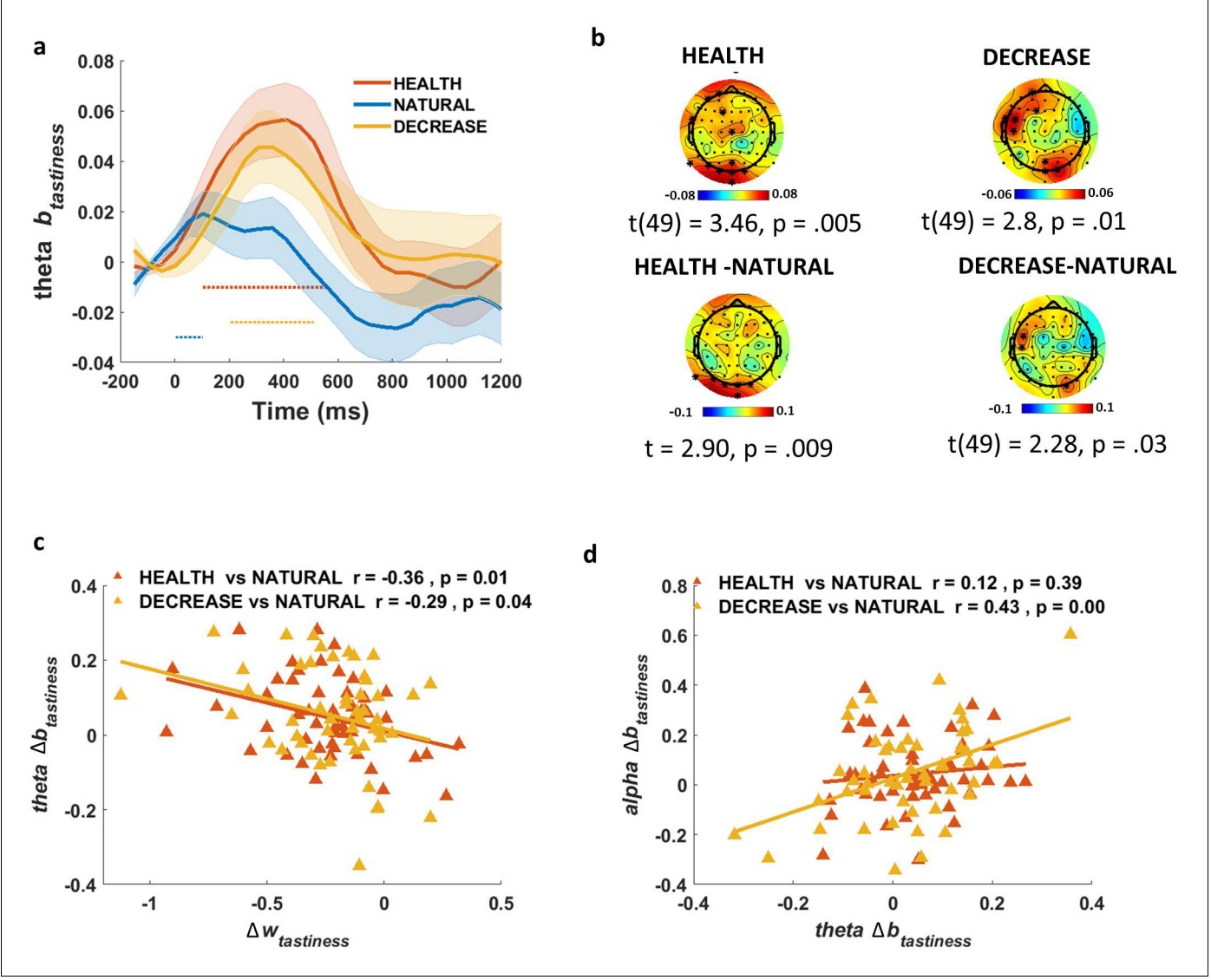

**Figure 5.** Theta-correlate of tastiness. (a) In the exploratory (not model-based) analysis, theta power coefficients for tastiness in Model 2 were significant on frontal and occipital channels in HEALTH and DECREASE conditions ~200–500 ms post-food.(b) Scalp distribution of the theta-correlate of tastiness in HEALTH (top left) and DECREASE (top right) and the difference in theta-correlate of tastiness in the HEALTH vs. NATURAL (bottom left) and DECREASE vs. NATURAL (bottom right) are shown. Theta-correlate of tastiness predicts (c) successful down-regulation of tastiness influence ($w_{tastiness}$) and is correlated with (d) alpha-correlate of tastiness across subjects; shaded error bars show within-subject standard error of the mean. Horizontal dotted lines show significant time bins (p<0.05).

in the ERP, predicted individual differences in successful down-regulation of the influence of tastiness on choice across two different regulatory strategies. Finally, we found an earlier rise in theta power at frontal and occipital areas of the scalp that did not correlate with model-predicted time courses but predicted regulatory success. Importantly, this signal correlated positively with food tastiness only when subjects regulated their decisions, and the more it did so, the greater the *decrease* in the influence of food tastiness on subjects' choices. This was true for both regulatory strategies. Taken together, our findings paint a nuanced portrait of the processes occurring during the regulation of value-based decision making. They suggest that regulation reduces the behavioural influence of tastiness not by disrupting early, stimulus-locked representations of tastiness, but by using a theta-related regulatory mechanism to disrupt the incorporation of tastiness into alpha-related EA processes.

## Functional significance of alpha suppression in EA

A main question concerns the processes that the alpha signal might represent in this context. One possibility is that this signal represents the final accumulated evidence for an integrated value signal that directly triggers a motor response. We think this is unlikely for several reasons. First and foremost, averaged alpha power failed to show the kind of clear rise to a fixed bound just before the time of response, as has been observed in both ERP and alpha power in motor cortex (*O'Connell et al., 2012*; *Steinemann et al., 2018*; *Twomey et al., 2015*). Second, the scalp distribution of alpha signals for tastiness and healthiness signals did not correspond perfectly, suggesting the possibility of distinct neural sources (although the poor spatial resolution prevents us from reaching a definitive conclusion). Finally, an integrated value signal should have shown robust sensitivity to both tastiness and healthiness according to the goals. Yet encoding of healthiness in alpha power was somewhat inconsistent. In health-sensitive channels, healthiness encoding occurred prior to the fastest decisions only in the HEALTH condition (*Figure 4—figure supplement 3g*). In taste-sensitive channels, healthiness encoding occurred during the slowest decisions only in the NATURAL condition (*Figure 4—figure supplement 3f*). Taken together, our findings suggest that the signals we observed might represent an intermediate stage of EA and leave somewhat ambiguous whether they are more consistent with an accumulation of an integrated evidence signal or instead accumulation of evidence about specific attributes. Further research, perhaps using combined EEG-fMRI, might help to resolve these important questions.

Beyond the question of what information this alpha signal represents, it is also important to consider the significance of this signal occurring in alpha oscillations, as opposed to other frequencies. Suppression of alpha oscillations, also called the event-related desynchronization of alpha, is widely known to reflect active information processing involving release from inhibition, perception, and attention and appears to be driven by cortico-cortical networks across frontal and parietal brain regions (*Foxe and Snyder, 2011*; *Klimesch, 2012*; *Klimesch et al., 2007*; *Samaha et al., 2018*). The degree of alpha suppression also predicts success in tasks where a higher level of cortical excitation is required (*Klimesch et al., 2007*), while inducing transient changes in alpha activity by external stimulation leads to predictable changes in cognitive performance (*Klimesch et al., 2003*). This literature strongly supports our observation that alpha suppression correlated with food attributes, that is, with tastiness when deciding naturally and with healthiness when focusing on health, and also correlates with behavioural evidence of regulatory success (*Figure 4a–d*).

Other work also supports the idea that prefrontal and posterior alpha suppression correlate with EA, although such studies have focused on perceptual decision making (*Kloosterman et al., 2019*; *van Vugt et al., 2012*; *Werkle-Bergner et al., 2014*). This work suggests that alpha suppression may relate to the control of cortical excitability in relation to accumulation of sensory evidence (*Kloosterman et al., 2019*) in line with the proposed role for alpha in regulation of inhibition: increase in alpha power (synchronization) reflects inhibition and decrease in alpha power (desynchronization) reflects release from inhibition (*Klimesch, 2012*). Yet the precise role of alpha suppression in representation of *value* during EA is not clear. Here, we found a striking similarity between the time course of alpha-correlates of tastiness and healthiness and the representation of these attributes in the simulated neural signals predicted by the best-fitting parameters of the DDM, particularly the EA signal (*Figure 4e, f*). One possibility, then, is that tastier foods may typically result in exactly this kind of release from inhibition, potentiating approach-related responding, but that this process is disrupted during regulation.

In this respect, it is interesting to speculate on the meaning of differences we observed between the relationship of alpha suppression to the influences of tastiness and healthiness on behaviour. While the *alpha-correlate of tastiness* showed a strong presence in occipital and frontal areas and correlated independently in both regulatory conditions with individual differences in down-regulation of tastiness, the *alpha-correlate of healthiness* was comparatively less consistent: it showed a more limited presence in posterior regions and did not correlate with up-regulation of healthiness across subjects. It is possible that this is simply due to lack of power to detect significant findings. Alternatively, we speculate that the HEALTH condition may activate both down-regulation of tastiness and up-regulation of healthiness. In combination with the relative *dominance* of the tastiness attribute in behaviour, this joint set of effects could result in a weaker *alpha-correlate of healthiness* or more variable dynamics in EA that are not fully captured by our model. For example, recent work

from *Maier et al., 2020* demonstrates complex effects of self-regulation on the dynamics of attention to healthiness and tastiness, which we have not incorporated in our model here. Future work will be required to tease apart these potential explanations. Overall, however, the close relationship between the DDM time course and alpha suppression indicates a quantifiable role for alpha suppression and its disruption during self-regulation.

Our results also indicated that alpha-correlates of tastiness and healthiness corresponded more closely with the predicted time course of EA, while ERP signals corresponded more closely to the predicted time course of attribute construction. Importantly, although alpha oscillations were sensitive to regulatory goal, ERP signals were not. This suggests that one function of alpha suppression may be to facilitate the translation of fronto-central attribute value signals into behavioural responses. When this alpha signal is disrupted, the influence of the frontal-central ERP on behaviour is likewise suppressed. Future work, perhaps focusing on inter-area neural synchrony and connectivity, might provide additional tests of this intriguing hypothesis.

## Functional significance of theta in self-regulation

Over the past several years, a consensus has emerged in the field that theta oscillations in the frontal cortex reflect cognitive control and integration of resources for cognitive control (e.g. *Cavanagh and Frank, 2014*; *Mas-Herrero and Marco-Pallarés, 2016*). Midfrontal theta, in particular, activates in response to any informative event or stimulus that conveys a need for behavioural adjustment or shift of attention (*Cavanagh and Frank, 2014*) and recruits regions underlying different cognitive operations to realize this need for control (*Mas-Herrero and Marco-Pallarés, 2016*). Posterior theta oscillations, on the other hand, may instead reflect the encoding of new information by the hippocampus in several memory operations (*Axmacher et al., 2010*; *Klimesch, 1999*; *Scholz et al., 2017*; *Staudigl and Hanslmayr, 2013*; *Tesche and Karhu, 2000*).

By contrast, evidence for the functional significance of the lateral frontal and posterior theta oscillations we observed in this task has not converged on a unified account. There is scattered evidence that early parietal and occipital theta oscillations correlate with value in high conflict choices (*Hunt et al., 2012*) and with cognitive load during mental calculation and multitasking (*Wang et al., 2018*). On the other hand, frontal theta in value-based decisions also shows sensitivity to emotional stimuli and efficient sensory integration in different studies (*Cavanagh et al., 2011*; *Nayak et al., 2019*; *Tosun et al., 2017*; *Zavala et al., 2016*).

Our findings may help to shed light on these inconsistencies. Notably, the *theta-correlate of tastiness* in our study appeared in both regulation conditions, but not unregulated decisions, around the same time, with a shared occipital component, and a similar though not identical frontal component (although the rough spatial resolution of EEG prevents us from making strong conclusions about whether such differences imply distinct neural sources). In other words, we observed stronger theta-correlates on precisely those trials which might be hardest to regulate (i.e. trials with tastier foods). This suggests that the theta-correlate of tastiness may represent an inhibitory control mechanism that helps to gate the impact of the constructed attribute that was reflected by front-central ERPS and prevent it from integrating into the processes reflected by alpha at frontal and posterior sites. Future studies that are more specifically targeted towards examining the relationships between ERPs, theta, and alpha oscillations, and deploy simultaneous EEG-fMRI to obtain better localization of the source of these signals, will be needed to confirm this idea. Brain stimulation techniques could further investigate whether disruption of early theta power causes impairments in regulation.

## Processes reflected by the ERPs and oscillatory power

As mentioned above, a striking finding of our study pointed to an important difference between the oscillatory and ERP signatures of attribute value construction and EA. While previous perceptual decision-making studies have found that ERPs, in particular the P300 at centro-parietal electrodes, reflect the EA process predicted by DDM (*O'Connell et al., 2012*; *Twomey et al., 2015*), our study found correlates of attribute values in ERPs with a more frontal distribution. These ERPs did not reflect the regulated integration of attributes to the EA process, instead coding for taste attribute values regardless of condition, consistent with prior work (*Harris et al., 2013*). To our knowledge, this is the first report to suggest a differentiation in the computational functions reflected by ERP and oscillatory signals during value-based choice, although such distinctions have been previously

reported in other cognitive processes (*Hajihosseini and Holroyd, 2013*; *Holroyd et al., 2012*). Future work, perhaps in conjunction with fMRI, will be needed to confirm this distinction and to tie it to different neural mechanisms.

## Comparisons to previous work on value and regulation

Our findings contribute to a growing but still scant EEG literature on the neural representation of components of value in the EA process. While some studies have examined modulation of EEG during self-control (*Harris et al., 2013*), and other studies have examined oscillatory correlates of EA itself (*Hunt et al., 2012*; *van Vugt et al., 2012*; *Werkle-Bergner et al., 2014*), ours is the first study to date to combine the two approaches. In this respect, it is informative to compare our results to previous work. For example, *Harris et al., 2013* reported an early attentional ERP component in occipital areas (N170) that was enhanced in successful self-control and later centro-parietal ERP components localized to the ventromedial prefrontal cortex that were sensitive to the overall value and tastiness and healthiness attributes. Similarly, in our study, we found two temporally distinct oscillatory signatures corresponding to valuation (alpha suppression around 600 ms post-stimulus) and regulation (theta enhancement around 200 ms post-stimulus). While a direct comparison of these two sets of findings is not possible due to differences in the study design and analysis, we speculate that the early EEG predictors of regulation observed in our study as well as in *Harris et al., 2013* might indicate a common oscillatory mechanism (occipital theta power) that controls the processing of attributes in the service of regulation. We also speculate that the later components of attribute value represented by alpha suppression in our study reflect the *controlled* integration of sensory evidence by areas involved in decision making and response selection.

Importantly, while we do observe neural signatures consistent with EA signals in our study, our results fail to replicate some previous reports in the literature finding a distinct gamma (46–64 Hz) signature of value-based EA (*Polanía et al., 2014*) at parietal and fronto-polar sites prior to response. Intriguingly, gamma phase coherence in this prior study revealed fronto-parietal connectivity during value-based decisions and not perceptual decisions. To test specifically whether we replicated this finding, we examined oscillatory power in the NATURAL condition averaged over all channels and specifically at Fz and Pz channels, but did not find an enhancement of gamma in relation to our pre-food 200 ms baseline (*Figure 4—figure supplement 4a*). To test whether this is due to higher variability in gamma phase in food-locked analysis, we also tested response-locked oscillatory power in relation to the pre-food baseline, but failed to find a significant increase in gamma power (*Figure 4—figure supplement 4b*).

We speculate that these discrepancies may be due to differences in the cognitive and memory demands presented by the different task paradigms. For example, our subjects make an accept/reject response to one food stimulus, whereas in *Polanía et al., 2014*, they responded by choosing one of two food stimuli presented on the upper or lower side of the screen while a scrambled version of each image was displayed by their side (Figure 1A in *Polanía et al., 2014*). This more complex paradigm might elicit higher frequency responses due to differences in visual and spatial processes. Alternatively, it is possible that the choice between two items vs. accepting or rejecting a single item, although behaviourally comparable in terms of the decision value, might require distinct working memory processes. Future work directly comparing different modes of choice (e.g. single-item vs. binary or multi-item choice) will be needed to resolve some of these questions.

In addition to gamma signals, *Polanía et al., 2014* found that fronto-central low-beta (18–20 Hz) power was also negatively correlated with EA. Similarly, we also observed suppression of low-beta power (*Figure 4—figure supplement 4*), though to a lesser degree compared to alpha power. A full match to the single-trial value-based predictions of our DDM model regarding EA only occurred with alpha suppression (*Figure 4e, f*) and did not match responses in the beta range. Notably, *Polanía et al., 2014* did not examine frequencies below 15 Hz or look for the kind of trial-by-trial variation in value that we identify here, making a direct comparison of our results to theirs difficult.

## Conclusion

Taken together, our results highlight the potential for model-based analyses to lend new insight into the neural mechanisms underlying self-regulation in value-based decision making. Our findings corroborate previous accounts suggesting that alpha suppression tracks EA and further suggest that

alpha suppression may represent an identifiable signature indicating the integration of value-based attributes into the EA process during both regulated and unregulated choices. By contrast, we found a representation of attribute values in ERPs that was not affected by regulation. Our study is also the first to our knowledge to find evidence that theta oscillations might play a role in inhibiting the influence of attribute values on EA, independent of regulation strategy. These findings point to the importance of understanding the distinct neural mechanism by which this inhibition occurs and may represent a target of future work to enhance the effectiveness of self-regulation in dietary choice.

## Materials and methods

### Subjects

We recruited 66 participants from the University of Toronto Scarborough research participation system and flyers posted on campus. Five subjects did not complete the task either by choice or due to technical issues. Eleven others were excluded based on our pre-registered exclusion criteria (more than three noisy channels in their EEG recording and/or if they made the same choice in more than 90% of trials in our *NATURAL* condition as described under *Task* [Open Science Framework; *HajiHosseini and Hutcherson, 2020*]). Participants were recruited until we obtained our pre-registered target sample size of 50 usable subjects (*HajiHosseini and Hutcherson, 2020*; https://osf.io/ewtvx/; 30 females, age range 17–31). All subjects gave written consent prior to the experiment. The study was approved by the Ethics Board of the University of Toronto. Subjects recruited through the research participation system received course credits while those recruited by flyers received $30 for participation.

### Task

To increase motivation to eat, all participants were instructed to refrain from eating for 3 hr before the start of the experiment. The experiment consisted of three tasks: a *pre-choice rating* task, a *self-regulation choice task (SRCT)*, and a *post-choice rating* task (*Hare et al., 2011*; *Harris et al., 2013*; *Hutcherson et al., 2012*). Subjects also reported how hungry they were on a scale of 1– 9 (hunger level) just before the pre-choice rating and after the SRCT. During the task, subjects were seated approximately 75 cm from a 24-inch computer screen.

#### Pre-choice rating task

In order to match the values of foods appearing in each condition, we recorded liking ratings for 208 different food stimuli before and after the SRCT. Subjects rated the foods on a scale of 1–6 corresponding to *strongly dislike*, *moderately dislike*, *slightly dislike*, *slightly like*, *moderately like*, and *strongly like*. Stimuli were presented in the centre of the screen filling 45% of the screen area. Rating keys were presented under the food picture. Based on these ratings, we selected 180 different foods and divided them into three, roughly equally liked sets of 60 foods each, for use in the SRCT task that followed.

#### SRCT

During the SRCT, participants made choices about whether to eat different food stimuli under three different conditions, presented in blocks of 15 trials. Sixty unique foods appeared in one of three conditions: focus on healthiness (HEALTH), decrease desire for food (DECREASE), or respond naturally (NATURAL). Subjects completed 36 randomly interleaved blocks (12 blocks of each condition). Each of the 60 food stimuli appeared three times throughout the blocks in a single condition (i.e. 180 trials per condition) in order to avoid confounding memory or value effects across conditions. At the beginning of each block, one of three sets of instructions corresponding to each condition was presented on the screen for 5 s.

> **NATURAL:** RESPOND NATURALLY; for the next set of trials, we would like you to RESPOND NATURALLY. Allow any feelings or thoughts you have to come naturally and make whatever choice you most prefer at that moment.
> **HEALTH**: FOCUS ON HEALTHINESS; for the next set of trials, we would like you to FOCUS ON THE HEALTHINESS OF THE FOODS. Consider how healthy the food is as you are deciding what you prefer to do. Then make whatever choice you most prefer at that moment.

**DECREASE:** FOCUS ON DECREASING DESIRE; for the next set of trials, we would like you to DECREASE YOUR DESIRE FOR FOOD. Do whatever you need to in order to decrease your craving for food as you decide what you prefer to do. Then make whatever choice you most prefer at that moment.

Every trial started with a fixation cross in the centre of the screen for 500 ms and was followed by a food stimulus. To minimize eye movements, the food stimulus was centred and filled only 20% of the screen. Subjects pressed one of four response keys (*d*, *f*, *j*, or *k*) on the keyboard to indicate their decision (*Strong No*, *No*, *Yes*, or *Strong Yes*) about each food, with right-left order of response counterbalanced across subjects. Participants had 4 s to decide. Following response, a randomly jittered 1–2 s inter-trial interval appeared. To ensure that subjects followed the correct instruction throughout every block, we signalled the conditions with a colour code: NATURAL = green, DECREASE = yellow, and HEALTH = red. A coloured frame appeared around the written instructions at the beginning of each block and the fixation cross at the beginning of each trial appeared in the colour corresponding to the condition (*Figure 1*).

## Post-choice attribute rating task

In addition to providing a second set of liking ratings for foods after the regulation task, subjects also rated the food stimuli from the SRCT for tastiness and healthiness on a scale of 1–6 (*very untasty* to *very tasty* and *very unhealthy* to *very healthy*). All foods were rated on one attribute, and then on the other, with rating order randomized across subjects. These subjective attribute ratings allowed us to assess how regulatory strategies altered the influence of attribute values on choice.

Following completion of liking, tastiness, and healthiness ratings, one trial from the SRCT was randomly selected for implementation and presented on the screen. The subject ate the food if they responded Yes or Strong Yes. Following completion of all experimental tasks, subjects completed a set of questionnaires designed to measure individual differences and real-world dietary behaviour.

We wrote the experiment codes in MATLAB using the Psychophysics Toolbox (*Brainard, 1997*).

## EEG

### Recording

We used a 64-channel BioSemi ActiveTwo system for recording the EEG during the SRCT, aligned according to the 10-20 system. We also recorded from three external channels, left and right mastoids for offline referencing and an electrode under the right eye to detect eye blinks. Recordings on Biosemi were sampled at 512 Hz, reference free, and later referenced to the averaged mastoids upon importing data to EEGLAB (*Delorme and Makeig, 2004*).

### Pre-processing

All channel data were high-pass filtered at 0.5 Hz to remove slow drifts and low-pass filtered at 100 Hz to remove high-frequency noise. A notch filter was applied to remove the 60 Hz line frequency. Data were then re-referenced to averaged channels in order to enable comparison with previous EEG studies of value-based decision making (*Harris et al., 2013*). We identified bad channels by visual inspection and replaced them by interpolation. To centre the data, average amplitude for each channel was subtracted from each data point on that channel. Data were then segmented from 1 s prior to the presentation of the food stimulus to 1.5 s after the presentation of the food stimuli. Average amplitude over a baseline window starting from 200 ms before the presentation of food stimulus was subtracted from all time points in the segment. Independent component analysis (ICA) was then performed on concatenated segments. Components associated with ocular and muscle artefacts or channel noise were identified based on their time and frequency information and removed. Channel data were reconstructed based on the remaining components and re-segmented into the original segments.

### Event-related potentials

Single-trial ERPs were constructed by segmenting pre-processed trials starting from 200 ms before to 1.5 s after the presentation of food stimulus at each channel. See *Pre-processing* for details on baseline correction.

## Time-frequency analysis

Each trial starting from 1 s before to 1.5 s after the presentation of food stimulus was convolved with a complex Morlet wavelet with a varying number of cycles (ranging from 3 for lower to 10 for higher frequency ranges) to study frequencies from 1 to 80 Hz in 1 Hz steps. The transformed time series contained a complex number for each time point, channel, and frequency, $Ae^{j\Phi t}$, where $A$ was the power and $\Phi$ the phase angle. We normalized the power in each time point ($A_t$) to average power at a baseline consisting of the 200 ms prior to onset of the food stimulus: $10 * \log_{10}(A_t/A_{baseline})$. We calculated changes in power on the single-trial level as explained in the Statistical analysis section below. We also averaged over the power of single-trial time-frequency data in all trials in each condition to assess the overall changes of power in each condition. Wherever results for fast, slow, and medium RT are reported, we split the trials based on 33rd and 66th percentile of RT for each subject separately and averaged over trials in each bin.

## Modelling overview

To investigate the computational bases of decisions, we fit a hierarchical multi-attribute drift diffusion model (HDDM) (*Wiecki et al., 2013*) to the choice and reaction time (RT) data from the SRCT using tastiness and healthiness ratings from the pre-rating task as predictors. We carried out the modelling in three stages. First, we performed basic estimation of model parameters using the HDDM package (*Figure 2—figure supplement 1*, *Figure 2—source data 1*). Second, we used the estimated parameter fits in our custom-written codes to simulate the time courses of three hypothesized neural signals during each trial: representation of the tastiness attribute construction process, representation of the healthiness attribute construction process, and representation of the EA process (*Figure 2a, b*). Finally, we estimated the effects of subjects' tastiness and healthiness ratings on these simulated signals using a series of regression analyses at each time point (*Figure 2c*). We describe each of these steps in more detail below.

## Drift-diffusion model (DDM)

The DDM included six parameters per condition: three parameters related to drift rate (i.e. evidence/value formation) in each trial (weights on tastiness, $w_{tastiness}$, healthiness, $w_{healthiness}$, as well as a constant *ValConst; total drift = $w_{tastiness}$ \* tastiness + $w_{healthiness}$ \* healthiness+ValConst*), and three additional parameters: decision threshold (*trs*), starting point bias (*spbias*), and non-decision time (*ndt*) representing post-stimulus perceptual and pre-response motor processes. HDDM uses hierarchical Bayesian estimation to identify the best-fitting parameters simultaneously for individual subjects and on the group level, assuming individual subjects' parameters are drawn from the group distribution. To estimate the joint posterior distribution of all parameters (individual and group parameters), we had HDDM sample 11000 iterations after an initial burn-in period of 1000 samples using Markov Chain Monte Carlo method. Best-fitting parameters were computed as the mean for that parameter averaged over the posterior distribution across all chains. We then used the HDDM posterior predictive check functions to inspect the posterior distribution of the parameters and the subject responses times (*Figure 2—figure supplement 1e*). We also found that the average simulated posterior predictive response times were significantly correlated with the observed response times across subjects and conditions (mean r = 0.46; t(49) = 17.1, p<0.001). Both the posterior predictive checks and the correlation analyses thus confirmed that the model provided a good fit to the data.

## Model-simulated neural signals

Following identification of best-fitting model parameters, we used the subject and condition-specific parameter values to simulate the expected time course of three signals on each trial: (1 and 2) the expected *attribute construction signals* related to tastiness and healthiness; and (3) the expected *EA signal* (*Figure 2a*). Each signal was computed from food onset to response time using individual subject parameters (*Figure 2b*). For signals related to attributes, the functions on each trial resembled a boxcar with onset immediately following termination of the perceptual part of the non-decision time (i.e. the estimated non-decision time parameter minus 80 ms to account for motor preparation time that occurs before response), offset immediately upon response, and a height equivalent to the attribute value on each trial (*Figure 2b*; blue and red lines). The EA signal, in contrast, accumulates

gradually from just after termination of the perceptual non-decision time until the response, with an average slope equal to the integrated value of the choice (*Figure 2b*; black lines). This approach thus allowed us to construct three simulated time courses: a predicted trial-by-trial time course for the neural representation of the tastiness attribute construction (taste-AC) signal, a predicted trial-by-trial time course for the neural representation of the healthiness attribute construction (health-AC) signal, and a predicted trial-by-trial time course for the integrated EA signal (*Figure 2a, b*). To produce a more stable prediction of the signals on each trial, we constructed 1000 simulated data-sets for each subject and took the average across these simulations. We performed this averaging step over multiple simulations because each individual simulation is slightly different due to the random noise in attribute and EA signal formation. Thus, averaging together a large number of simulations produces more stable results. These simulated signals represent hypothetical time courses that are based on fundamental assumptions of the DDM and test to what extent these assumptions capture particular aspects of EEG dynamics.

## Attribute contributions to model-simulated neural signals

Finally, we asked how these signals might appear in neural data when locked to onset of the food. For example, to determine the expected time course of a neural signal representing the tastiness attribute (i.e. taste-AC), within each condition (i.e. NATURAL, HEALTH, DECREASE), we computed a separate regression for each time point $t = 0$ to 2000, (i.e. from food onset to 2000 ms afterwards), using the average model-simulated trial-by-trial taste-AC signal at that time point as the dependent variable, and the subject's trial-by-trial tastiness and healthiness ratings, as well as RT, as predictors. We included RT in the model to control for the possible shared variance between attribute ratings and RT in the attribute construction and EA signals, so that the model coefficients are only driven by strength of evidence and not their secondary effect on RT. We applied this same procedure to determine effects of tastiness, healthiness, and RT on the simulated healthiness signal as well as the simulated EA signal (Model 1).

$$
\begin{aligned}
taste/health - &\ AC \ or \ EA(t) \\
\cong b(t)_0 + b(t)_{tastiness} &* tastiness + b(t)_{healthiness} * healthiness \\
&+ b(t)_{rt} * rt
\end{aligned}
\qquad \text{Model 1}
$$

The resulting $b(t)_{tastiness}$ and $b(t)_{healthiness}$ coefficients at each time point represent when and to what extent we would expect to observe an association between (1) the tastiness ratings and a neural signal computing tastiness, (2) healthiness ratings and a neural signal computing healthiness, and (3) tastiness and healthiness ratings and a neural signal performing a process of weighted EA, over and above the association of these ratings with RT and with each other. The distinct time course of these simulated coefficients for each condition can be seen in *Figure 2c*. We used these model-predicted time courses to identify EEG signals conforming to model predictions of either attribute or EA signals (see below for details). For brevity, we omit a discussion of results related to $b(t)_{rt}$.

## Statistical analysis

### Behaviour and DDM

We analysed behavioural data both on the subject and trial level. We used three-way ANOVAs with NATURAL, HEALTH, and DECREASE as within-subjects factors to study the effect of regulation on RT, choice, and DDM parameters across subjects. Post-hoc paired t-tests were used to compare regulation conditions with NATURAL and with each other.

### EEG

For comparison to simulated computational time courses derived from time point by time point regressions (see above), we performed regression analyses on the single-trial ERPs where amplitude on each trial was averaged across each 49 ms time bin (resulting in 25 data points) for all channels. Similarly, we performed regression analysis on time-frequency power data, dividing the average power of frequencies from 1 to 80 Hz into six frequency bands: delta (1–3 Hz), theta (4–8 Hz), alpha (9–12 Hz), beta (13–35 Hz), gamma1 (36–59 Hz), and gamma2 (61–80 Hz). Time-frequency power (*tfPower*) was averaged across each 49 ms time bin (25 data points) for all channels. We then regressed trial-by-trial *ERP*s and *tfPower* in every frequency band onto the trial-by-trial *tastiness* and

*healthiness* of the food item presented on each trial and on *RT*. This allowed us to assess the time course by which trial-by-trial variation in each attribute was related to ERPs and tfPower, controlling both for RT and the other attribute (Model 2).

$$ERP/tf\ Power(t)$$
$$\cong b(t)_0 + b(t)_{tastiness} * tastiness + b(t)_{healthiness} * healthiness$$
$$+ b(t)_{rt} * rt$$

Model 2

Using this model, we extracted $b(t)_{tastiness}$ and $b(t)_{healthiness}$ for ERPs and each frequency band power, at every time bin, and channel for each subject separately in every condition, allowing us to construct an empirically observed time course of beta coefficients that was analogous to the model-predicted time courses estimated from Model 1 (see above). In addition to these analyses using all trials, we conducted similar analyses dividing trials into the fastest and slowest half of response times. Where we report results for fast and slow RTs, we split the trials based on the 50th percentile of RT per subject and ran Model 2 for each set of trials separately.

## Comparison of EEG time course to simulated DDM time course

To identify ERP and oscillatory signals where these time courses corresponded to predicted signals from the DDM, we regressed time courses of $b(t)_{tastiness}$ and $b(t)_{healthiness}$ from Model 2 (i.e. EEG regressions) across each channel and each frequency band (for time-frequency power) onto their counterparts from Model 1 (i.e. simulated DDM regressions) separately for the taste attribute construction, health attribute construction, and EA signals. This allowed us to scale the time course of ERP and time-frequency power to the simulated neural data. Finally, we calculated the correlation between the scaled ERP and time-frequency power time course and the simulated DDM time course from 0 to 1.2 s following stimulus presentation ($r_{eeg-model}$). This allowed us to examine the extent to which the time course of attribute-related components of EEG resembled predictions of the computational model. We compared the correlation between model and EEG time courses for attributes and EA using a method for comparing dependent correlations (*Lenhard and Lenhard, 2014*).

We retained only those channels (for ERPs) and channel × frequency bands (for tfPower) with joint significance criterion where (1) $r_{eeg-model}$, that is,. the correlation between *b*-coefficients of EEG from Model 2 and *b*-coefficients of taste, health-AC, or EA signals from Model 1 was >0.85 and significant at p<0.05, Bonferroni corrected for number of comparisons (64 channels × 7 EEG components: six frequency bands and the ERP) *and* where Model 2 *b*-coefficients of ERP or power in the corresponding frequency were significantly different from zero (df = 49, p<0.05) across participants within a contiguous window of ±100 ms around the peak of averaged model predictions for a given signal (i.e. taste attribute construction, health attribute construction, or EA signal), with a window length cutoff at 1000 ms post-food stimulus.

Note that this approach allows us to compare the time course of ERP and oscillatory dynamics in the decision period against the time course of model simulations that integrate all model parameters in order to study the temporal dynamics of attribute and EA representations in the EEG. We employed the joint criterion for significance to ensure that the model and EEG peak latencies are aligned before correlation across time is calculated.

Additionally, in *post hoc* exploratory analyses, we examined the time course of *b*-coefficients of Model 2 for effects that did not follow model-predicted temporal trajectories. For this analysis, we retained any channels with at least six consecutive time bins (i.e. > 300 ms) in which *b*-coefficients of power or ERPs were significantly different from zero (df = 49, p<0.05) across participants with a window length cutoff at 1000 ms post-food stimulus.

## Additional information

### Funding

| Funder | Grant reference number | Author |
| --- | --- | --- |
| Natural Sciences and Engineering Research Council of Canada | RGPIN-2016-05641 | Cendri A Hutcherson |

| Canada Research Chairs | Cendri A Hutcherson |
|---|---|
| Connaught Fund | Cendri A Hutcherson |

The funders had no role in study design, data collection and interpretation, or the decision to submit the work for publication.

### Author contributions

Azadeh HajiHosseini, Conceptualization, Software, Formal analysis, Validation, Investigation, Visualization, Methodology, Writing - original draft, Project administration, Writing - review and editing; Cendri A Hutcherson, Conceptualization, Resources, Software, Supervision, Funding acquisition, Validation, Methodology, Writing - review and editing

### Author ORCIDs

Azadeh HajiHosseini (iD) https://orcid.org/0000-0001-7621-6527
Cendri A Hutcherson (iD) http://orcid.org/0000-0002-4441-4809

### Ethics

Human subjects: All subjects gave written consent for data collection and publication prior to the experiment. The study was approved by the Research Ethics Board of the University of Toronto (Protocol #34322).

### Decision letter and Author response

Decision letter https://doi.org/10.7554/eLife.60874.sa1
Author response https://doi.org/10.7554/eLife.60874.sa2

## Additional files

### Supplementary files

• Transparent reporting form

### Data availability

Raw data are deposited on Open Science Framework, under the project DOI: 10.17605/OSF.IO/EWTVX. Raw Behavioural data: https://osf.io/yp2x9. Raw EEG data: https://osf.io/p5wd2.

The following dataset was generated:

| Author(s) | Year | Dataset title | Dataset URL | Database and Identifier |
|---|---|---|---|---|
| HajiHosseini A, Hutcherson C | 2021 | EEG Dynamics of Self-Regulatory Strategies in Dietary Decision Making | https://doi.org/10.17605/OSF.IO/EWTVX | Open Science Framework, 10.17605/OSF.IO/EWTVX |

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
