## [Decision Letter]

**Acceptance summary:**

This work by HajiHosseini and Hutcherson investigates the self-regulation of choices about whether to accept or reject different varieties of food. Participants made choices either according to their natural preference, with an emphasis on healthiness, or by intentionally decreasing their desire for food. Choice behaviour across these three contexts was modelled via the drift diffusion model, and decision-relevant signals from the model were correlated with frequency-specific signals in scalp EEG data. The study will be of interest to researchers studying value-based decision making, mathematical models of decision making and their electrophysiological correlates.

**Decision letter after peer review:**

Thank you for submitting your article "Alpha and theta oscillations contribute to attribute regulation in dietary decision making under self-control" for consideration by *eLife*. Your article has been reviewed by 2 peer reviewers, and the evaluation has been overseen by a Reviewing Editor and Michael Frank as the Senior Editor. The following individuals involved in review of your submission have agreed to reveal their identity: James Cavanagh (Reviewer #1); Peter R Murphy (Reviewer #2).

The reviewers and reviewing editor have discussed the reviews with one another and the Reviewing Editor has drafted this decision to help you prepare a revised submission.

Both reviewers expressed enthusiasm for the overarching rationale of this paper and agree that it addresses an important topic that would be of considerable interest to the *eLife* readership. However, as you will see, the two reviewers also identified a number of very substantial issues relating to the general clarity of the manuscript and the informativeness of the analysis approach which may limit the extent to which firm conclusions can really be drawn regarding the mechanisms underlying value-based decision-making. Below you will see that the reviewers have suggested a considerable number of revisions that we would deem necessary before the paper could be considered for publication. If you are willing to complete these revisions, and/or provide a detailed rebuttal where relevant, the paper will be subjected to a further detailed second-round review. As the editors have judged that your manuscript is of interest, but as described below that substantial revisions are required before it is published, we would like to draw your attention to changes in our revision policy that we have made in response to COVID-19 (https://elifesciences.org/articles/57162). First, because many researchers have temporarily lost access to the labs, we will give authors as much time as they need to submit revised manuscripts. We are also offering, if you choose, to post the manuscript to bioRxiv (if it is not already there) along with this decision letter and a formal designation that the manuscript is "in revision at *eLife*". Please let us know if you would like to pursue this option. (If your work is more suitable for medRxiv, you will need to post the preprint yourself, as the mechanisms for us to do so are still in development.)

Summary:

This work by HajiHosseini and Hutcherson investigates the self-regulation of choices about whether to accept or reject different varieties of food. Participants made choices either according to their natural preference, with an emphasis on healthiness, or by intentionally decreasing their desire for food. Choice behaviour across these three contexts was modelled via the drift diffusion model, and decision-relevant signals from the model were correlated with frequency-specific signals in scalp EEG data. The main findings are that suppression of alpha-band power during decision formation was correlated with tastiness and healthiness of food items in contexts where these attributes were most relevant to the choice, which is partially consistent with model-predicted evidence accumulation signals; and, post-stimulus theta power selectively encoded food tastiness in contexts where the influence of this attribute on choice was suppressed

Essential Revisions:

1. Manuscript Clarity and Missing Details

(A) There are major issues with clarity throughout the report. This is primarily due to a lack of description of how the multiple stages of modeling were performed, but there are stylistic deficits as well.

(B) Lines 82-85: This should be a section of stating competing hypotheses, but it is hard to understand how these ideas – as described here – differ from each other and how they could be tested. This is a missed opportunity for providing additional clarity.

(C) The lack of display for things like RT and choice per condition, as well as DDM parameters is another missed opportunity for enhanced clarity (1/2 of pg. 10, most of pg. 11). Figure 2c is particularly frustrating to not know the average RT for each condition. There is currently little information provided by which to judge the quality of the model fits. All that is provided is in Figure S1 – but how the figure should be interpreted is unclear, and from what I can tell it provides no indication as to how well the model fits RT distributions (as opposed to just mean RT and choice percentages).

(D) On line 263, it details how the parameter was the mean of the posterior distribution, but it isn't clear if this individual level distribution is in any way constrained by the group level distribution, which would help protect against over-fitting.

(E) The fitted parameters need to be reported (and ideally, plotted). Of particular importance, currently there is no indication of the signs or magnitudes of the fitted weights given to the tastiness and healthiness attributes, only analyses of differences in weights.

(F) I would welcome a better explanation of how the weights given to different attributes are i) applied to the subject-specific food ratings, and ii) combined to yield a single drift rate for a given stimulus.

(G) It is not clear at all how the DDM params, taste and health params, and EEG are all fit together. Figure 2 caption details how these attribute estimates are "based on DDM structure", but it isn't clear how these boxcars are integrated into the EA signal, or how they are fit together to yield the curves in Figure 2c. The authors note that there were 1000 simulated datasets, but what varied between them?

(H) The cross correlations between parameters should be detailed at every level of the model. Co-linearity would be a major issue to avoid, and it seems inevitable in some cases like comparing attribute regressions to EA regressions (where EA is based on some combination of attributes). Without resolution of this issue, Figure panels 3d-e are not interpretable.

2. Analysis Approach

(A) The authors' chosen approach to identify decision signals in the EEG data is indirect and in my opinion of limited value. The issue here is using a correlation of model-estimated (Figure 2c) and observed (Figure 3a,b) signal trajectories to make inferences about the possible functional characteristics of the neural signals in question: When the model-estimated trajectories for both 'decision evidence' and 'evidence accumulation' (EA) signals are themselves highly similar, then they will necessarily yield highly similar correlations with any neural signal (in the case of the alpha suppression signal, both |r|>0.9) and provide very limited insight about what the neural signal might reflect. A more fruitful approach would surely be to exploit the fundamental differences that are predicted of evidence and EA signals: in particular the fact that, in the model at least, the former are expected to be essentially static during decision formation while the latter are expected to exhibit several dynamical properties (in build-up rate scaling with evidence strength, peak latency tracking response time, and stereotyped amplitude at response) that clearly demarcate the two types of signal. Identifying the latter properties in the alpha suppression signal would therefore provide far more compelling support for the proposal that this signal reflects evidence accumulation.

(B) Interpretation of alpha suppression effects. The above point is also relevant when considered in light of existing knowledge about alpha suppression during decision formation. Many studies have reported alpha suppression in the post-stimulus, pre-response period of decision-making tasks, though this response is rarely directly identified with the evidence accumulation process itself; rather, alpha suppression, at least over posterior scalp, tends to be identified with a kind of gating or release from inhibition process (including in papers cited by the authors). If such a process is in some sense sensitive to RT (which has been observed with alpha suppression), then it would perhaps not be surprising if it showed the characteristics observed for alpha suppression presently in Figure 3a,b (assuming RT correlates especially strongly with tastiness in the NATURAL condition and healthiness in the HEALTH condition, both of which are I believe supported by the behavioural modelling). In short, I think the meaning of the alpha suppression findings is currently unclear.

(C) In both Figures 3 and 4, the topographic distributions of the reported effects are by and large quite different between the different experimental conditions. This casts doubt on the notion that the reported signals reflect the same neural processes subject to contextual modulations, and raises further questions as to how the different results should be interpreted.

(D) Modelling. Was the 6-parameter model tested against any other model parameterizations? This can be critical for not over-fitting the data with params that aren't beneficial (e.g. starting point bias), This issue dovetails with the lack of clarity concerns noted above: starting point tends to soak up variance due to asymmetrical thresholds, suggesting both thresholds are enhanced during choice, but the RT distributions aren't shown so this remains unknown. It would be great to see these distributions so the difference between attribute vs. neutral distributions could be visually assessed for skew vs. kurtotic changes. Overall there are quite many free parameters (18) being fit to a relatively low number of trials (540; 180 per condition) per subject, and the model fits may suffer from overfitting. I would encourage the authors to fit more constrained models and identified the most parsimonious fit via model comparison.

(E) What is the purpose of the DECREASE condition? The instructions for this condition struck me as being very much open to interpretation, and indeed it seems to have led to some counterintuitive results (an increase in the weight given to healthiness in this condition, and generally increased decision bounds). Generally, if the instruction is to 'decrease my desire for food', why don't I just refuse every food item I am presented with?

Minor Revisions:

(A) It is surprising that the introduction doesn't include O'Connell's work on P3b slope and drift rate (line 95) (e.g. The classic P300 encodes a build‐to‐threshold decision variable, EJN). With the estimated value attributions peaking around 542 ms, this seems like it might relate to the slope to the P3b. Of course, this is all conjecture since neither the RTs nor the ERPs are shown, but I suggest that the authors utilize the similar correlation strategy with the raw EEG to see if it corresponds with known ERP component activities that have previously been linked to these same DDM constructs.

---

## [Author Response]

Essential Revisions:1. Manuscript Clarity and Missing Details(A) There are major issues with clarity throughout the report. This is primarily due to a lack of description of how the multiple stages of modeling were performed, but there are stylistic deficits as well.

We appreciate that the reviewer brings these issues to our attention. In the revised manuscript, we have broken the description of our modeling into 3 sub-sections, starting at lines 681, 698, and 721 of the clean version of the revised manuscript and have provided additional details in text and figure captions to improve the clarity of method descriptions. We have also addressed the lack of clarity in other parts as raised in the comments below.

(B) Lines 82-85: This should be a section of stating competing hypotheses, but it is hard to understand how these ideas – as described here – differ from each other and how they could be tested. This is a missed opportunity for providing additional clarity.

We thank the reviewer for this helpful comment. We have now explained the predictions made by these alternative hypotheses in lines 85-91.

“If the former is true, regulation should directly influence the neural representation of an attribute such as tastiness (perhaps early on in the decision process) whereas if the latter is true, the neural representation of an attribute might remain intact and only the later stages of decision making, when the attribute is incorporated into evidence accumulation, should change. Unfortunately, fMRI studies (Hare et al., 2009; Hutcherson et al., 2012) cannot dissociate the neural representations of attributes and evidence accumulation with the necessary temporal resolution for testing these distinct processes”.

We have also added a summary of how our results partially provide evidence for these hypotheses in lines 120-125.

“Our results suggested that the time course of ERPs correlated more strongly with model-based predictions for attribute representations, while suppression of alpha oscillatory activity correlated better with model-based predictions for evidence accumulation signals. Importantly, regulation had a larger effect on the alpha-correlate of tastiness compared to the ERP-correlate of tastiness, and this effect appeared to be mediated by an increase in the power of theta oscillations in the early stages of decision formation.”

and 382-386.

“Taken together, our findings paint a nuanced portrait of the processes occurring during the regulation of value-based decision making. They suggest that regulation reduces the behavioural influence of tastiness not by disrupting early, stimulus-locked representations of tastiness, but by using a theta-related regulatory mechanism to disrupt the incorporation of tastiness into alpha-related evidence accumulation processes.”

(C) The lack of display for things like RT and choice per condition, as well as DDM parameters is another missed opportunity for enhanced clarity (1/2 of pg. 10, most of pg. 11). Figure 2c is particularly frustrating to not know the average RT for each condition. There is currently little information provided by which to judge the quality of the model fits. All that is provided is in Figure S1 – but how the figure should be interpreted is unclear, and from what I can tell it provides no indication as to how well the model fits RT distributions (as opposed to just mean RT and choice percentages).

To address these shortcomings in the revised manuscript, we show the DDM parameter values and the distribution of RT for data and the model for each condition in supplementary material (Figure 2—figure supplement 1, Figure 2- source data 1). We have also marked the graphs in Figure 2c with average RT in each condition. In addition, we now report the posterior predictive checks in the main manuscript on lines 691-696 suggesting that the model generally fits the data well.

“We then used the HDDM posterior predictive check functions to inspect the posterior distribution of the parameters and the subject responses times (Figure 2—figure supplement 1e). […] Both the posterior predictive checks and the correlation analyses thus confirmed that the model provided a good fit to the data.”

(D) On line 263, it details how the parameter was the mean of the posterior distribution, but it isn't clear if this individual level distribution is in any way constrained by the group level distribution, which would help protect against over-fitting.

In the original version of the manuscript, we had fit the DDM parameters to individual subjects separately without constraints on the group level. However, we agree with the reviewer that a hierarchical model with group parameter constraints can assure more reliable estimates in terms of individual vs group fits. We have now revised our model and fit a 6-parameter hierarchical DDM using the HDDM package (Figure 2—figure supplement 1, source data 1). The HDDM package applies group constraint on the individual parameter estimates. Details are provided on line 686-691. Author response image 1 shows the individual DDM parameters calculated by the hierarchical method plotted against those calculated by our original method. As you can see, the fits are very similar across the two methods, and lead to nearly identical results in all analyses.

**Author response image 1. sa2fig1:** Comparison between DDM parameters calculated through simple and hierarchical MCMC algorithms.

(E) The fitted parameters need to be reported (and ideally, plotted). Of particular importance, currently there is no indication of the signs or magnitudes of the fitted weights given to the tastiness and healthiness attributes, only analyses of differences in weights.

We apologize for this neglect. We have revised Figure S1 (currently Figure 2—figure supplement 1) to reflect the mean and standard error of the mean for each parameter (Figure 2—figure supplement1 a-d). We also include a supplementary table (Figure 2- source data1) with details of mean and standard error for the group parameters. We have not reported the mean of parameters in text for brevity but would be happy to do so if reviewers and/or editor prefer.

(F) I would welcome a better explanation of how the weights given to different attributes are i) applied to the subject-specific food ratings, and ii) combined to yield a single drift rate for a given stimulus.

We estimated the overall drift rate on each trial as the weighted sum of the food attributes and a constant value: *v* = *w_tastiness_*tastiness*+ *w_healthiness_*healthiness*+*ValConst*. The magnitude of the attributes for *tastiness* and *healthiness* were provided by subject ratings and *w_tastiness_*, *w_healthiness_*, and *valConst* parameters were obtained by the DDM model fitting for each subject. We now explain this in more detail in the Modeling section of the Methods section (lines 681 to 686) and Figure 2 caption.

(G) It is not clear at all how the DDM params, taste and health params, and EEG are all fit together. Figure 2 caption details how these attribute estimates are "based on DDM structure", but it isn't clear how these boxcars are integrated into the EA signal, or how they are fit together to yield the curves in Figure 2c. The authors note that there were 1000 simulated datasets, but what varied between them?

We fit the DDM parameters to behaviour (i.e., response and RT) using subjects’ tastiness and healthiness ratings as predictors of the drift rate on each trial. After fitting and verifying with posterior predictive checks that these parameters provided a reasonable fit to the data (Figure 2—figure supplement 1e), we used the individual fitted parameters to create the 3 predicted time courses, for each subject, for each trial. These time courses represented simulated tastiness and healthiness attribute construction (AC) signals, as well as simulated evidence accumulation (EA) signals. We used the attribute weights and intercepts (combined with subject-specific health and taste ratings) to create the simulated drifts on each trial, using non-decision time and threshold to simulate the onset and offset of the evidence accumulation process, and starting point bias to simulate its starting value (lines 698-713; Figure 2b). We used 1000 simulations of each trial, with random Gaussian noise added to the attribute signals at each time point and carried forward into the accumulation signal (line 713-719). We then took the average of these simulations over all 1000 simulations to construct a single predicted time-course on each trial, for each subject. Then we conducted regressions with the average trial-by-trial simulated attribute construction or evidence accumulation signal at each time point as the dependent variable, and healthiness, tastiness, and RT on each trial as predictors, in order to find the predicted contribution of these attributes to the simulations of taste and health attribute constructions (AC) and evidence accumulation (EA) signals (Figure 2c; lines 721-742). Finally, we averaged the resulting coefficients across subjects to produce the final predicted group-level *b(t)_tastiness_* and *b(t)_healthiness_* time courses for each of taste and health AC, and EA signals.

We then regressed the group-averaged regression coefficients from Model 2 (regression of ERP or oscillatory power on tastiness, healthiness, and RT) at each channel on these time courses and reconstructed the projected *b(t)_tastiness_* and *b(t)_healthiness_* time courses based on the regression coefficients (lines 756-768). We retained channels where our criteria were met (lines 783-791):

“We retained only those channels (for ERPs) and channel × frequency bands (for tfPower) with joint significance criterion where 1) *r_eeg-model_* (i.e., the correlation between *b*-coefficients of EEG from Model 2 and *b*-coefficients of taste, health-AC, or EA signals from Model 1 was >.85 and significant at p <.05, Bonferroni corrected for number of comparisons (64 channels × 7 EEG components: 6 frequency bands and the ERP) *and* where Model 2 *b*-coefficients of ERP or power in the corresponding frequency were significantly different from zero (df = 49, p <.05) across participants within a contiguous window of +/- 100 ms around the peak of averaged model predictions for a given signal (i.e., taste attribute construction, health attribute construction, or evidence accumulation signal), with a window length cut-off at 1000 ms post-food stimulus.”

(H) The cross correlations between parameters should be detailed at every level of the model. Co-linearity would be a major issue to avoid, and it seems inevitable in some cases like comparing attribute regressions to EA regressions (where EA is based on some combination of attributes). Without resolution of this issue, Figure panels 3d-e are not interpretable.

We thank the reviewer for raising this point. We have taken a number of approaches to address it.

First, the reviewer is correct that the correlation between the representation of tastiness/healthiness in the attribute construction time-courses and evidence accumulation time courses is quite high (Pearson’s r ranging from.88 to.96 across all conditions). To address this issue, we used a method described in Lenhard and Lenhard (2014) for comparing the strength of correlation between related/dependent measures to determine whether the attribute construction or EA signals showed a higher correspondence to the observed EEG time course of the ERP and alpha-correlates of tastiness and healthiness (lines 224-227, 274-279, and 321-324; Figure 3b, and 4e,f).

In addition, based on other suggestions in this review, we sought to control for other correlated variables such as RT. Following their advice, we assessed the correlation between RT and tastiness and healthiness ratings using *RT ~ b0 + b_tastiness_*tastiness + b_healthiness_*healthiness* which confirmed their prediction that RT and tastiness are indeed moderately correlated (Author response image 2).

**Author response image 2. sa2fig2:** Contribution of tastiness (left) and healthiness (right) to RT in all conditions.

To control for this shared variance, we have revised our Model 1 and Model 2 to include RT as a predictor, so that the tastiness and healthiness contributions to the simulated signals and EEG are calculated over and above their effect on RT and on each other (line 728-734 and 756-765).

Importantly, controlling for RT actually strengthened our conclusions, showing that after accounting for collinearity, the correlation between model and attribute construction time courses in ERP data |r_erp-AC_| is significantly larger than the correlation with the evidence accumulation time course |r_erp-EA_| (Figure 3b). In contrast, the correlation between alpha power and the model-predicted time course of evidence accumulation |r_alpha-EA_| is significantly larger than the correlation with the predicted time course of attribute construction signals |r_alpha-AC_| (Figure 4e-f).

Finally, in supplementary analyses described below (see response to Essential revisions #2A, point 2, Author response image 3), we examined the match between the rate of change/derivative of tastiness representations in alpha power and the rate of change/derivative of the tastiness representations in the simulated AC/EA signals, since the time course of the derivatives for attribute and evidence accumulation time-courses are less correlated than the main signal. Although this analysis showed a reduced correlation between the tastiness correlates of alpha slope and simulated signals (|Pearson’s r| ranging from.41 –.71) compared to our original analysis (which we think is likely due to noise), it confirms a significantly stronger relationship between the time course of trial-by-trial alpha power and the EA signal than with the attribute construction signal. In combination, we hope that these analyses show that our findings are robust across different ways of analyzing the data, and all confirm the idea that the correlates of alpha power that we observe more closely match the expected EA signal.

**Author response image 3. sa2fig3:** a) time course of contribution of tastiness to alpha slope of change (red line) plotted against time course of contribution of tastiness to slope of simulated tastiness attribute construction (AC, dashed black line) and slope of simulated evidence accumulation (EA, dotted black line) signals, b) time course of contribution of healthiness to alpha slope of change (blue line) plotted against time course of contribution of healthiness to slope of simulated healthiness attribute construction (AC, dashed black line) and slope of simulated evidence accumulation (EA, dotted black line) signal; Shaded error bars show within-subject standard error of the mean.

2. Analysis Approach(A) The authors' chosen approach to identify decision signals in the EEG data is indirect and in my opinion of limited value. The issue here is using a correlation of model-estimated (Figure 2c) and observed (Figure 3a,b) signal trajectories to make inferences about the possible functional characteristics of the neural signals in question: When the model-estimated trajectories for both 'decision evidence' and 'evidence accumulation' (EA) signals are themselves highly similar, then they will necessarily yield highly similar correlations with any neural signal (in the case of the alpha suppression signal, both |r|>0.9) and provide very limited insight about what the neural signal might reflect. A more fruitful approach would surely be to exploit the fundamental differences that are predicted of evidence and EA signals: in particular the fact that, in the model at least, the former are expected to be essentially static during decision formation while the latter are expected to exhibit several dynamical properties (in build-up rate scaling with evidence strength, peak latency tracking response time, and stereotyped amplitude at response) that clearly demarcate the two types of signal. Identifying the latter properties in the alpha suppression signal would therefore provide far more compelling support for the proposal that this signal reflects evidence accumulation.

We thank the reviewer for this comment and suggestions.

1) We agree with the reviewer that isolating the effects of attributes on attribute construction and evidence accumulation signals based on the correlation of time courses faces challenges. However, it is important to note that our aim in this study was to highlight the strengths of model-based analysis of EEG oscillations. Therefore, we designed our analysis steps based on model predictions of the time course dynamics (simulated signals) as opposed to selection of a priori time windows and channels to test for specific decision parameters. The hypothesis-driven parts of our analyses were the fundamental assumptions of DDM about the integration of attribute values with the evidence accumulation process (the boxcars and EA signals in Figure 2b). This is the reason why we employed the joint significance criterion (lines 783-791) that assures the latency of maximum contribution of attributes to ERP and alpha (the peak and trough of the waveform in Figures 3a and 4a,b respectively) occurs +/- 100 ms around the peak latency contribution of attributes to the simulated AC and EA signals (Figure 2c) at channels where we found the ERP-correlate of tastiness and alpha-correlates of tastiness and healthiness. In fact, our revised analysis shows that based on these criteria, alpha power correlates of tastiness and healthiness are significantly closer to the model predictions of the taste-EA signal compared to model predictions of the taste-AC signal (z = 4.09, p<.001) and health-AC signal respectively (z = 4.17, p<.001; lines 274-276 and 321-324; Figure 4e,f), whereas ERP data are significantly closer to the model predictions of the taste-AC signal compared to the taste-EA signal (z = 2.55, p=.005; lines 224-227; Figure 3b).

2) In addition, we followed the reviewer’s advice and investigated the relationship between attributes and the slope/build-up rate of change in alpha power by applying Model 2 to the derivative of alpha power at each time point. To parallel our original simulation, we did this by simulating the derivative/instantaneous slope of contributions of attributes to tastiness and healthiness attribute construction and EA signals and compared the model simulation against data (derivative of power) at channels that we identified as encoding alpha correlates of tastiness and healthiness (marked channels in Figure 4 c, top-row). We found that time courses of attribute contribution to alpha power derivative/slope are also correlated with their model counterparts although this analysis is, perhaps not surprisingly, slightly noisier, resulting in slightly lower correlations. Nevertheless, these analyses also show a striking correspondence between the model-predicted time course of evidence accumulation and the observed signal (Author response image 3). In addition, this correlation is significantly stronger for the evidence accumulation compared to the attribute construction signal for tastiness (|r_taste(alpha-EA)_|>|r_taste(alpha-AC)_ |; z = 1.82, p=.03), although it fails to reach significance for healthiness (p>.05).

3) Based on the reviewer’s advice, we also studied whether the peak latency of alpha suppression, averaged across trials in the NATURAL condition, in channels where we found the alpha-correlate of tastiness tracks RT in trials with slow, medium, and fast response times. We found that alpha power in fast trials reached its trough earlier than the slower trials (Author response image 4) bearing resemblance to the previous findings that argued P300 carried an evidence accumulation signal (O’Connell, Dockree, and Kelly, 2012; Twomey, Murphy, Kelly, and O’Connell, 2015). As the signal of our interest was in fact contribution of attribute rating to alpha power (like b_tastiness_) in these channels, we followed up by running Model 2 on slow and fast trials separately, which we think is a cleaner and more rigorous test of our hypotheses than simply averaging signal over all trials; Note that we used only 2 response speed bins to avoid inadequate number of observations in the regression model. The results (Author response image 4, B) showed that our alpha-correlate of tastiness indeed scaled with RT, lasting longer for slow trials compared to fast trials (all conditions split by fast and slow RTs are shown in Figure 4—figure supplements 1). By contrast, the same analysis on the ERP-correlate of tastiness did not show such relationship with response time (Author response image 4, D). These results lend extra support to our interpretation that the alpha-correlate of tastiness represents evidence accumulation whereas the ERP-correlate of tastiness represents a choice-insensitive attribute construction process.

**Author response image 4. sa2fig4:** Alpha-correlates of tastiness and healthiness in NATURAL condition for fast (a) and slow (b) trials; ERP-correlates of tastiness and healthiness for fast (c) and slow (d) trials.

Although we think that the taste-correlation by RT analyses are more sensitive, we also performed the exact test requested by the reviewer and studied whether averaged alpha power in the same channels reached a stereotyped amplitude prior to response across slow and fast trials. We found that alpha power started to build up between 400 to 200 ms before the response, and reached a peak at roughly 200-300ms before the response. However it did not reach a clearly unique bound across slow and fast trials (Figure 4—figure supplement 2B). These results suggest that the signal we observe is likely not the final evidence accumulation leading directly to the motor response (which we suspect resides in motor circuits), but rather an intermediate stage of evidence accumulation displaying some but not all functional properties of the modelled DDM signal.

Interestingly, when we ran our Model 2 on response-locked alpha power separately for fast and slow trials, we found the predicted sensitivity to tastiness right before the response was present for both trial types (Figure 4—figure supplement 3E, F). However, we also found that in slow trials, the alpha sensitivity to tastiness followed an elongated pattern, and further showed some more significant sensitivity to healthiness (Figure 4—figure supplement 3F). In health-sensitive channels on slow trials, in contrast, alpha sensitivity to healthiness is weaker and farther from the response time (Figure 4—figure supplement 3G, H). We suspect these results might indicate arbitration or oscillation between attention to attributes that is not explicitly modeled in the standard drift diffusion model we employ here, but that has been shown to be potentially important in other work from our own labs and others (e.g., Maier, Raja Beharelle, Polanía, Ruff, and Hare, 2020; Sullivan, Hutcherson, Harris, and Rangel, 2015). This oscillation in attention likely creates more variability in the neural signal across trials, perhaps contributing to the pattern we see in Figure 4—figure supplement 3F, H. We are currently working on developing more sophisticated models of attribute sampling based on these ideas, and hope to use the current data to test those ideas. However, we think they are beyond the scope of the current paper, and so have chosen to continue using the more canonical DDM to model our results.

Taking stock of these results together, we think that they suggest that our alpha-correlate of tastiness represents accumulation of evidence but perhaps at an intermediate level before the final motor response. Therefore it is affected by the variability and distribution of response times and regulation across trials, but does not show the full pattern of expected responses if it is the final stage of evidence accumulation for response selection. We have thus now added some of these speculations in the Results (lines 290-315) and Discussion (lines 388-405) sections of the revised manuscript.

In the Results section, we write:

“Next, we conducted a further test of the idea that the alpha-correlate of tastiness relates to evidence accumulation, by examining response-locked data in fast and slow decisions. […] Such architecture would be consistent with observations from other work finding evidence for multiple evidence accumulation mechanisms working at different stages to determine the final choice (e.g., Steinemann, O’Connell, and Kelly, 2018).”

And in the Discussion section, we now write:

“A main question concerns the processes that the alpha signal might represent in this context. […] Further research, perhaps using combined EEG-fMRI, might help to resolve these important questions.”

(B) Interpretation of alpha suppression effects. The above point is also relevant when considered in light of existing knowledge about alpha suppression during decision formation. Many studies have reported alpha suppression in the post-stimulus, pre-response period of decision-making tasks, though this response is rarely directly identified with the evidence accumulation process itself; rather, alpha suppression, at least over posterior scalp, tends to be identified with a kind of gating or release from inhibition process (including in papers cited by the authors). If such a process is in some sense sensitive to RT (which has been observed with alpha suppression), then it would perhaps not be surprising if it showed the characteristics observed for alpha suppression presently in Figure 3a,b (assuming RT correlates especially strongly with tastiness in the NATURAL condition and healthiness in the HEALTH condition, both of which are I believe supported by the behavioural modelling). In short, I think the meaning of the alpha suppression findings is currently unclear.

Following the reviewer’s point, we tested the correlation between RT and tastiness and healthiness in different conditions and revised our regression models to control for this shared variance across attribute ratings and RT (see response to Essential revisions #1H). Including RT in Model 1 and Model 2 ensures that our results on the contribution of attribute values to oscillatory power and simulated time courses are not driven by RT. Notably, we continue to find effects of tastiness and healthiness that correlated with model-predicted time courses (Figure 4e,f), independent of and controlling for RT. We thus think that this is unlikely to explain our effects.

In addition to the effect of regulation, in the revised manuscript, we also tested the alpha-correlate of tastiness for slower and faster responses and found that this correlate was scaling with RT so that it lasted longer in slow trials compared to fast trials (Author response image 4, B). We believe that these findings together provide more clarification and support to our interpretation of the alpha-correlate of tastiness.

(C) In both Figures 3 and 4, the topographic distributions of the reported effects are by and large quite different between the different experimental conditions. This casts doubt on the notion that the reported signals reflect the same neural processes subject to contextual modulations, and raises further questions as to how the different results should be interpreted.

We agree with the reviewer that the distribution of these effects is not fully consistent across the different conditions and attributes. This issue pertains both to the alpha and theta oscillations.

For alpha signals, we now acknowledge this issue in the discussion, lines 393-395 and 432-436. Other than possible lack of statistical power, we consider some possibilities for this lack of consistency. In Figure 4c (Figure 3 in the original manuscript), we present the correlates of tastiness and healthiness in the NATURAL and HEALTH conditions respectively. It is possible that the representations of tastiness and healthiness are actually different; tastiness is perhaps perceived through available food features whereas healthiness might be inferred according to features and a priori knowledge. This could lead to differences in the neural encoding of these attributes, or their representation in evidence accumulation signals. In fact, our revised interpretation of the alpha-correlates of tastiness and healthiness is that they likely present an intermediate stage of integration of attribute representations to the evidence accumulation signal and therefore might be driven by distinct sources (see lines 290-315 in the Results section and lines 388-405 in the Discussion section for details on how we describe this in the manuscript).

For theta signals, the evidence is a bit less clear as we acknowledge in lines 465-466 of Discussion. In Figure 5b (Figure 4 in the original manuscript), we present the theta- correlate of tastiness that is associated with suppression of tastiness in HEALTH and DECREASE conditions. While there is some overlap in the sensors that code for tastiness in the two conditions, there are also distinct topographies. We think these differences may be due to the distinct cognitive processes occurring in the two conditions, which might result in recruitment of both overlapping and distinct regulatory mechanisms. For example, suppression of tastiness by theta might be a more general process in DECREASE as subjects are instructed to decrease their desire for food independent of its tastiness and healthiness, while in the HEALTH condition, this suppression might be a by-product of focusing on healthiness. In sum, we think that the difference in distributions of alpha-tastiness and healthiness correlates and theta-tastiness correlate across the two regulation conditions may be driven by real neural differences. We have acknowledged this more thoroughly in the Discussion section on lines 437-446 and 472-484, and point to the need for further work (in particular, the added value of simultaneous EEG-fMRI versions of this paradigm) for resolving these issues.

(D) Modelling. Was the 6-parameter model tested against any other model parameterizations? This can be critical for not over-fitting the data with params that aren't beneficial (e.g. starting point bias), This issue dovetails with the lack of clarity concerns noted above: starting point tends to soak up variance due to asymmetrical thresholds, suggesting both thresholds are enhanced during choice, but the RT distributions aren't shown so this remains unknown. It would be great to see these distributions so the difference between attribute vs. neutral distributions could be visually assessed for skew vs. kurtotic changes. Overall there are quite many free parameters (18) being fit to a relatively low number of trials (540; 180 per condition) per subject, and the model fits may suffer from overfitting. I would encourage the authors to fit more constrained models and identified the most parsimonious fit via model comparison.

We thank the reviewer for raising this concern. We have now included the RT distribution in Figure 2—figure supplement 1 (see response to Essential revisions #1C). Regarding the number of parameters, we included the 3 regression parameters (w_tastiness_, w_healthiness_, ValConst) as they were essential to explore our question of interest (i.e. changes in value as a function of regulation). Threshold and non-decision time were also essential to form the DDM model traces. Following the reviewer’s suggestion, we also fitted a different model that included the 5 parameters mentioned above but excluded the starting point bias parameter. The average DIC across the three conditions was 14878 for the 5-parameter model and 14520 for the 6-Parameter model indicating the better fit of the 6-parameter model. We have not reported this comparison in the manuscript in order not to interrupt the flow but we can alternatively include it in the supplementary material if reviewers find it necessary. Also, note that in HDDM, it is not possible to fit a model *without* a value constant. However, in the previous version of our paper, using custom-written scripts, we also found that models that did not include a constant performed considerably worse. Thus, we believe the model we present is the minimum model required to fully account for the data.

We also note that the number of parameters we use to fit each condition is consistent with other published work in value-based decision making, but with an even larger number of trials. For example, we refer the reviewer to previous research from our own lab e.g., Hutcherson, Bushong, and Rangel (2015-*Neuron*), Tusche and Hutcherson (2018- *eLife*), as well as others e.g., Maier et al. (2020- *Nature Human Behaviour*) where models with a similar number of parameters have been used with similar task designs.

(E) What is the purpose of the DECREASE condition? The instructions for this condition struck me as being very much open to interpretation, and indeed it seems to have led to some counterintuitive results (an increase in the weight given to healthiness in this condition, and generally increased decision bounds). Generally, if the instruction is to 'decrease my desire for food', why don't I just refuse every food item I am presented with?

The reason for including this condition is two-fold. First, we followed up on previous studies which have used this more general approach to regulating food craving (e.g., Kober et al., 2010; Hutcherson et al., 2012). Second, an appealing aspect of this condition is that it decreases the tastiness weight without necessarily changing the healthiness weight. Thus, it provides a useful comparison to the HEALTH focused condition, because it reduces the taste weight using an alternative strategy, while having comparatively smaller impact on the healthiness weight. We have made the logic for the inclusion of this condition clearer in the introduction (lines 105-111). Note also that participants clearly did *not* simply refuse all foods in this condition, which we believe likely results from our instruction to participants to always indicate their true preference after regulating, even if that was inconsistent with the instruction, and with the fact that this was an incentivized task in which, if the participant said no to everything, they would go hungry during the 30-minute waiting period after completion of the study. This instruction, combined with the behavioural results and model fits (which indicate a good fit to behaviour in this condition) suggest that our participants were indeed continuing to evaluate foods in this condition, and were partially successful in decreasing their desire for foods. They did this largely by reducing the effect of tastiness on their choices while not increasing the effect of healthiness as strongly. Note also that the small increase in healthiness weight is actually not wholly counterintuitive: a large portion of the food stimuli used in this study were unhealthy foods, so increasing the health weight might naturally reduce the overall value of most of these foods.

Minor Revisions:(A) It is surprising that the introduction doesn't include O'Connell's work on P3b slope and drift rate (line 95) (e.g. The classic P300 encodes a build‐to‐threshold decision variable, EJN). With the estimated value attributions peaking around 542 ms, this seems like it might relate to the slope to the P3b. Of course, this is all conjecture since neither the RTs nor the ERPs are shown, but I suggest that the authors utilize the similar correlation strategy with the raw EEG to see if it corresponds with known ERP component activities that have previously been linked to these same DDM constructs.

We apologize for this neglect. We have now included the reference. In the previous version of the manuscript, we had mainly focused on the oscillations as there was less clarity in the literature about their role in attribute value construction and evidence accumulation. Following the reviewer’s comment, we tested stimulus-locked ERPs in correlation with our models. While we did not find a correlate of healthiness that meets our significance criterion, we found frontal and central (CPz, FC1) and parietal (P7, P9, P10) channels where contribution of tastiness to ERPs was continuously significant ~400-700 ms post food stimulus and the time course of the simulated contribution of tastiness to tastiness signal was correlated (>.85) with the ERPs (Figure 3). We found that strikingly, this ERP-correlate of tastiness matched closely with the tastiness attribute signal as opposed to the EA signal (z = 2.55, p = .005; Figure 3b). This finding was very valuable in relation to our research question as it provides complementary evidence on the existence of distinct neural signatures of attribute value construction and EA in the EEG signal. Therefore we decided to re-organize the manuscript in order to report these findings. We want to thank the reviewer for this suggestion, since we think it has made the manuscript considerably more interesting and informative.

**References**:

Hutcherson, C., Plassmann, H., Gross, J. J., and Rangel, A. (2012). Cognitive regulation during decision making shifts behavioral control between ventromedial and dorsolateral prefrontal value systems. The Journal of Neuroscience: The Official Journal of the Society for Neuroscience, 32(39), 13543–13554. https://doi.org/10.1523/JNEUROSCI.6387-11.2012

Lenhard, W., and Lenhard, A. (2014). Hypothesis tests for comparing correlations. Psychometrica.

Maier, S. U., Raja Beharelle, A., Polanía, R., Ruff, C. C., and Hare, T. A. (2020). Dissociable mechanisms govern when and how strongly reward attributes affect decisions. Nature Human Behaviour, 1–15. https://doi.org/10.1038/s41562-020-0893-y

O’Connell, R. G., Dockree, P. M., and Kelly, S. P. (2012). A supramodal accumulation-to-bound signal that determines perceptual decisions in humans. Nature Neuroscience, 15(12), 1729–1735. https://doi.org/10.1038/nn.3248

Sullivan, N., Hutcherson, C., Harris, A., and Rangel, A. (2015). Dietary self-control is related to the speed with which attributes of healthfulness and tastiness are processed. Psychological Science, 26(2), 122–134. https://doi.org/10.1177/0956797614559543

Tusche, A., and Hutcherson, C. A. (2018). Cognitive regulation alters social and dietary choice by changing attribute representations in domain-general and domain-specific brain circuits. https://doi.org/10.7554/*eLife*.31185.001

Twomey, D. M., Murphy, P. R., Kelly, S. P., and O’Connell, R. G. (2015). The classic P300 encodes a build-to-threshold decision variable. European Journal of Neuroscience, 42(1), 1636–1643. https://doi.org/10.1111/ejn.12936